# THE SCENE LANGUAGE: REPRESENTING SCENES WITH PROGRAMS, WORDS, AND EMBEDDINGS

## ABSTRACT

We introduce the Scene Language, a visual scene representation that concisely and precisely describes the structure, semantics, and identity of visual scenes. The Scene Language represents a scene with three key components: a program that specifies the hierarchical and relational structure of entities in the scene, words in natural language that summarize the semantic class of each entity, and embeddings that capture the visual identity of each entity. This representation can be inferred from pre-trained language models via a training-free inference technique, given text or image inputs. The resulting scene can be rendered into images using traditional, neural, or hybrid graphics renderers. Together, this forms a robust, fully automated system for high-quality 3D and 4D scene generation. Compared with existing representations like scene graphs, our proposed Scene Language generates complex scenes with higher fidelity, while explicitly modeling the scene structures to enable precise control and editing. Project page: https://sclg-page.github.io/.

## 1 INTRODUCTION

How do you describe a scene? Imagine that you just traveled to Easter Island and would like to explain to Alice the wondrous scene of Ahu Akivi: "There are seven moai in a row, facing the same direction." "What is a moai?" Alice asked. "A moai is a stone human figure without legs, but each of them also looks slightly different." At this point, you realize it seems difficult to precisely explain the scene using natural language alone.

In fact, this example highlights a complete scene representation requires at least three types of complementary information: (1) *structural knowledge*, which is about the joint distribution of multiple instances, like "seven moai in a row, facing the same direction," most naturally described as programs; (2) *category-level semantics*, which may be shared across instances, often described in words, such as "moai"; (3) *instance-level intrinsics*, tied to the identity of each specific object or part, such as its geometry, color, and texture, which is hard to describe but easy to recognize.

Modern AI techniques provide natural grounding for each of the three modalities, while also falling short of capturing all: in-context learning of pre-trained language models (LMs) enables the inference of domain-specific programs (Brown et al., 2020); LMs capture rich semantic information based on words in natural language; embeddings obtained via techniques like textual inversion (Gal et al., 2023) or low-rank adaptation (Hu et al., 2021) best capture object identity. However, none of these existing representations alone is sufficient for scene generation and editing.

We introduce the Scene Language, a representation that integrates the three modalities—programs, words, and embeddings—to precisely and concisely describe the structure, semantics, and identity of visual scenes. In the Scene Language, a program specifies a computation process that defines the organization of a collection of *entities* in the scene, including extrinsics like poses and structural regularity like repetitions. Each entity is associated with a word referring to its semantic group, as well as an embedding describing its instance-specific attributes.

In addition to the representation itself, we propose a training-free inference module using a pre-trained LM as a backbone to infer the Scene Language from texts and images. When provided with a domain-specific language (DSL) for scenes, LMs decompose the task of complex scene generation into simpler tasks of scene component generation by predicting their corresponding modular functions. We also discuss possible neural, traditional, and hybrid graphics engines that render the

Figure 1: **Structured Scene Generation and Editing Using the Scene Language.** We develop a scene representation for 3D scene generation and editing tasks. Given textual scene descriptions, the representation can be inferred by a pre-trained large language model, rendered in 3D, and edited following language instructions. The representation contains a program consisting of semantic-aware functions bound to words, providing high interpretability and an intuitive scene-editing interface, and embeddings enabling editing with fine controls, *e.g.*, transferring the style of $<z1*>$ from a user-input image to the generated scene.

representation to images. Together, the Scene Language, the inference module, and the renderer lead to a robust system for high-quality, detailed 3D and 4D scene generation and editing.

In summary, our contributions are as follows.

1. A scene representation, the Scene Language, capturing structure, semantics, and identity of visual scenes using programs, words, and embeddings.

2. A training-free method that infers the representation from texts and/or images using pre-trained language models.

3. A generic rendering module that renders the Scene Language into an image.

4. Empirical results on text- and image-conditioned scene generation and editing tasks.

## 2 RELATED WORK

Visual scene representations are arguably the most fundamental problem in computer vision; thus, for sure, we may not enumerate all related work. As our Scene Language comprises programs, words, and embeddings, we will organize our discussion accordingly into three categories: scene representations that use program-based representations (Section 2.1), semantic graph-based representations (Section 2.2), and a pre-trained generative model's latent space (Section 2.3).

### 2.1 REPRESENTING SCENES AS PROGRAMS

Programs can specify not only the relations among scene components mentioned in Section 2.2, but also structural patterns such as hierarchy and repetitions, making them suitable as explicit descriptions of scene structures. Prior works have proposed to use programs in the form of sequences of execution commands as object-centric representations, followed by neural executors that render the programs into 3D shapes (Tian et al., 2019; Sharma et al., 2018; Deng et al., 2022). In comparison, ShapeAssembly (Jones et al., 2020) introduces higher-level functions with semantically meaningful function names, *e.g.*, "chair" and "back", to its program representation. Both ShapeAssembly and ours adopt the design principle of function abstraction, which results in clearly stated hierarchy relation among components and better program editability. However, ShapeAssembly uses cuboids as the shape representation and does not model appearance, while ours allows for more precise geometry and appearance modeling using expressive neural embeddings.

All the representations mentioned above require 3D datasets for training. More recently, with the advance of language models (LMs), several methods (Zhou et al., 2024b; Hu et al., 2024; Yamada et al., 2024; Sun et al., 2023; Zhang et al., 2023a; Tam et al., 2024) have proposed to use zero-shot LM inference for generating programs that will be rendered into scenes. These methods operate on

| Section 3.1 | Section 3.2 | Definition |
|---|---|---|
| *Operations* | | |
| $\Psi_{\text{transform}}$ | `transform` | Transform an entity |
| $\Psi_{\text{union}}$ | `union` | Compose entities |
| $f_w : z \mapsto h$ | `entity-func` | Entity function mapping embedding to entity |
| | `primitive-func` | Entity function mapping embedding to primitive |
| $f_w(z)$ | `(call word embedding)` | Function evaluation |
| *Data Types* | | |
| $w$ | `Word` | Word describing semantics |
| $t$ | `Matrix` | Entity pose |
| $z$ | `Embedding` | Embedding specifying entity identity |
| $h$ | `Entity` | An entity |
| $s$ | | The represented scene |

Table 1: **Summary of Notations in Sections 3.1 and 3.2.**

top of program syntax from specific graphics renderers such as Blender[1], and they do not permit parameters in high-dimensional embedding spaces unlike ours.

## 2.2 REPRESENTING SCENES WITH SEMANTIC GRAPHS

Prior semantic scene representations often adopt a graph to encode semantic scene components, such as objects and parts. In particular, Yuille & Kersten (2006); Huang et al. (2018) propose to employ a parse graph of context-free grammar, using terminal nodes to correspond to objects and their attributes, to represent a scene. Both works employ an analysis-by-synthesis approach to infer the representation from images that heavily rely on domain-specific priors. Alternative representations include scene graph (Johnson et al., 2015; 2018; Gao et al., 2024), where each node in a graph corresponds to an object and an edge corresponds to a pairwise relation, and StructureNet (Mo et al., 2019), which focuses on an object-centric setting and uses nodes for object parts. While these representations preserve the high-level semantics of scenes or objects, they leave out low-level precision; thus, geometric, textural, or relational details that cannot be fully specified by language or hand-crafted rules are often ignored. We address this issue via the inclusion of embeddings.

## 2.3 REPRESENTING SCENES WITH GENERATIVE MODEL LATENTS

The latent space of visual generative models can serve as a representation space for visual scenes. Such latent space can effectively capture the exact visual content of scenes, including geometry and appearance details, and can be either directly inferred, *e.g.*, in variational inference (Kingma, 2013) and model inversion (Zhu et al., 2016). More recently, text-to-image diffusion models have shown remarkable results in image synthesis. This class of models offers several candidate representation spaces including the space of textual embeddings (Gal et al., 2023), low-rank network weights (Hu et al., 2021), full model weights (Ruiz et al., 2023), or noise vectors in the diffusion process (Song et al., 2021; Mokady et al., 2023; Ho et al., 2020). However, such representations typically do not offer interpretable semantics or explicitly encode hierarchical scene structures. We incorporate textual embeddings into our structural representation in this work, leveraging its high expressivity to preserve visual details.

## 3 THE SCENE LANGUAGE

We aim to design a visual scene representation that encodes the structure, semantics, and visual content of scenes. Towards this goal, we propose the Scene Language, which represents a scene with three components: a program that encodes scene structure by specifying the existence and relations of scene components, which we will refer to as entities; words in natural language that denote the semantic group of each entity in the scene; and neural embeddings that pertain the low-level visual details and identities of the entities by permitting an expressive input parameter space. In the following, we will first give a formal definition of the representation (Section 3.1), and then introduce a domain-specific language (DSL) (Section 3.2) as its realization.

### 3.1 FORMAL DEFINITION

The Scene Language for a scene $s$, denoted as $\Phi(s)$, is formally defined as follows:

$$\Phi(s) := (P, W, Z). \tag{1}$$

---

[1] https://www.blender.org/

```
(bind "chessboard"
  (lambda (<z_>)
    (union
      (call "board" <z_>)
      (call "chess pieces" <z_>))))

(bind "chess pieces"
  (lambda (<z_>)
    (define (create-pawns i)
      (let* ((white-pawn (call "pawn" <z_>))
             (black-pawn (call "pawn" <z_>))
             (white-pose (translate (list (* i 0.125) 0.02 0.125)))
             (black-pose (translate (list (* i 0.125) 0.02 0.75)))
             (white-pawn-t (transform white-pawn white-pose))
             (black-pawn-t (transform black-pawn black-pose)))
        (union white-pawn-t black-pawn-t)))
    (define pieces-order
      '("rook" "knight" "bishop" "queen" "king"
        "bishop" "knight" "rook"))
    (define (create-other-pieces i)
      (let* ((piece-order (list-ref pieces-order i))
             (white-piece (call piece-order <z_>))
             (black-piece (call piece-order <z_>))
             (white-pose (translate (list (* i 0.125) 0.02 0)))
             (black-pose (translate (list (* i 0.125) 0.02 0.875)))
             (white-piece-t (transform white-piece white-pose))
             (black-piece-t (transform black-piece black-pose)))
        (union white-piece-t black-piece-t)))
    (union (union-loop 8 create-pawns)
           (union-loop 8 create-other-pieces))))

(bind "pawn"
  (lambda (<z_>)
    (transform (primitive-func <z_>) (scale (list 0.1 0.1 0.1)))))

;; ... (other binding expressions)
```

Figure 2: **Scene Language Overview**. A Scene Language represents a scene with three components: a program consisting of entity functions, a set of words (*e.g.*, ``pawn''') denoting the semantic class of the entity functions, and a list of embeddings (*e.g.*, <z1>) capturing the identity of each entity in the scene. Each entity function is bound with an entity class name given by a word, and maps an input embedding to an output entity of that class. Executing the program effectively computes all entities; the computation graph is shown on the right. Entity dependency, as indicated by arrows, reflects the hierarchical relation of entities in a scene. See Section 3.1 for representation definitions and Section 3.2 for program syntax. The program shown is converted from our inference method output, with text prompt "a chessboard at game start"; raw outputs in Appendix G.1.

Here, $P := \{f_w\}_{w \in W}$ is a program consisting of a set of entity functions $f_w$, where each entity function $f_w$ defines a class of entities in the scene, such as "board" and "pawn" illustrated in Fig. 2 and is uniquely identified by such a word, *e.g.*, $w =$ "board" in natural language, which succinctly summarizes its semantic meaning. $W$ denotes the collection of words corresponding to all the entity functions in the program. Each entity function $f_w$ maps a neural embedding $z$ to a specific entity $h$ in the scene, where $z$ specifies the attributes and identity of the output entity, like a specific color of a "pawn". Hence, the complete Scene Language $\Phi(s)$ of a particular scene $s$ also contains a list of neural embeddings $Z := [z_1, z_2, \cdots, z_J]$ encoding $J$ specific entities $[h_1, h_2, \cdots, h_J]$ in the scene.

Crucially, the program $P$ captures scene structures in three aspects. First, each entity function $f_w$ in $P$ transforms and composes multiple sub-entities (*e.g.*, 64 squares) into a new, more complex entity (*e.g.*, board), naturally reflecting the hierarchical, part-whole relations in the scene, as the arrows in Fig. 2 highlight. Second, multiple entities $h_j$ in the scene may belong to the same semantic class $w$ (*e.g.*, "square"), and can thus be represented by reusing the same entity function $f_w$ with distinct embeddings $z_j$. Finally, each entity function also captures the precise spatial layout of the sub-entities by specifying their relative poses during the composition, such as 64 squares forming an $8 \times 8$ grid.

In the following, we will expand on how functions from $P$ are defined, followed by the program execution procedure to compute the represented scene $s$. Notations are summarized in Table 1.

**Entity Function Definitions.** An entity function $f_w : z \mapsto h$ maps an embedding $z$ to an entity $h$, and $h$ is said to have an identity specified by $z$ and belongs to a semantic class $w$. Specifically, to obtain an entity $h$, $f_w$ is applied recursively:

$$h = f_w(z; \Omega(z)) := \Psi_{\text{union}}(\Psi_{\text{transform}}(h^{(1)}, t^{(1)}), \cdots, \Psi_{\text{transform}}(h^{(N)}, t^{(N)})),$$
$$\text{where} \quad h^{(i)} = f_{w^{(i)}}(z^{(i)}; \Omega(z^{(i)})), \quad i = 1, 2, \cdots, N, \tag{2}$$

| Data Types | |
|---|---|
| `Word` | // Word specifying semantics |
| `Embedding` | // Embedding specifying intrinsic attributes |
| `Vector` | `::= Array[Float]` // Vector in $\mathbb{R}^3$ |
| `Matrix` | `::= Array[Array[Float]]` // Transformation in $\mathrm{GA}(3, \mathbb{R})$ |
| `Entity` | `::= List[Entity]` // Recursively defined |
| | `\| Tuple[Word, Embedding, Matrix]` |

| Grammar | |
|---|---|
| `entity-func` | `::= (lambda (_::Embedding) create-entity)` |
| `create-entity` | `::= (call word embedding)` |
| | `\| (primitive-func embedding)` |
| | `\| (union create-entity create-entity)` |
| | `\| (union-loop loop-count loop-func)` |
| | `\| (transform create-entity matrix)` |
| `loop-func` | `::= (lambda (_::Integer) create-entity)` |
| `word` | `:: Word` |
| `embedding` | `:: Embedding` |
| `loop-count` | `:: Integer` |
| `matrix` | `:: Matrix` |

| Macros | |
|---|---|
| `primitive-func ::` | `Embedding -> Entity` // Create a primitive entity |
| `union ::` | `Entity -> Entity -> Entity` // Compose entities |
| `union-loop ::` | `Int -> (Int -> Entity) -> Entity` // Compose entities with for loop |
| `transform ::` | `Entity -> Matrix -> Entity` // Transform entity pose |

| Special Forms | |
|---|---|
| `(bind <word> (lambda (<formal params>) <body>))` // Defines and binds a function |
| `(call <word> <actual params>)` // Calls a function identified by word |

Table 2: **The Domain-Specific Language.** The following table contains the DSL specification used to define our representation. Built-in data types (*e.g.*, `String`), special forms (`lambda`, `define`, `let`, `let*`), and conditionals (`if`) are omitted. `::=` denotes definition; `::` denotes type annotation; `_::` denotes type annotation for an anonymous function formal parameter.

and $\Omega(z) = \{z^{(1)}, z^{(2)}, \cdots \}$ retrieves the list of embeddings corresponding to its sub-entities. Here, $\Psi_{\text{transform}}$ transforms a sub-entity $h^{(i)}$ with a pose $t^{(i)}$, and $\Psi_{\text{union}}$ composes multiple sub-entities $h^{(i)}$ into one single entity $h$. Each sub-entity $h^{(i)}$ is computed from another entity function $f_{w^{(i)}}$ by applying Eq. (2) recursively. For instance, let $f_w$ denote the entity function that produces the board in Fig. 2 (namely, $w = $ "board"). This function $f_w$ composes 64 sub-entities $h^{(i)}, i = 1, 2, ..., 64$ of the same class "square", which are in turn obtained by executing the *same* entity function $f_{w^{(i)}} = f_{\text{"square"}}$ with *different* embeddings $z^{(i)}$.

**Program Execution.** To obtain a scene $s$ from the Scene Language $\Phi(s) = (P, W, Z)$, a program executor identifies a root entity function $f_{w_1}$ from $P$ that is not dependent by any other function (*e.g.*, $w_1 = $ "chessboard" from Fig. 2), and evaluates this root function using the first element of the embeddings $z_1 \in Z$ to obtain $s = f_{w_1}(z_1)$. Evaluating $f_{w_1}(z_1)$ expands the computation recursively to its children functions $h_j = f_{w_j}(z_j)$ as defined in Eq. (2), obtaining a full sequence of all the entities $h_j$ of the scene, where $j = 2, 3, \cdots, J$, embedding $z_j \in Z$, and word $w_j \in W$. An example of the expanded computation graph is visualized on the right of Fig. 2.

## 3.2 THE SCENE LANGUAGE AS A PROGRAMMING LANGUAGE

We now concretize the definition in Section 3.1 with a domain-specific language (DSL) specified in Table 2. To define entity functions in the DSL, we introduce macro operations `union` for $\Psi_{\text{union}}$, `union-loop` which calls `union` on entities evaluated in a for-loop, and `transform` for $\Psi_{\text{transform}}$. We further include `primitive-func` in the DSL, which implements a primitive entity function that only depends on itself (*i.e.*, no children). We use these four macro operations and function calls of dependent functions to define entity functions. In particular, we allow variable assignment in the function body (*e.g.*, `let*` and `define` in Fig. 2). Entity functions are identified with the associated words in the DSL via two special forms: `bind`, which binds an entity function $f_w$ to word $w$, and `call`, which retrieves $f_w$ given $w$.

```
[<z1>,<z2>,...,<z9>]

(bind "scene" ...)
(bind "moai" ...)
(bind "base" ...) ...
```

Program Execution →

```
[[("moai",<z2>,t2),
...,
("moai",<z8>,t8)],
("base",<z9>,t9)]
```

**Graphics Renderer**

$g_{\text{reparam}}$ → $\left((\theta_1, t_1), (\theta_2, t_2), \cdots\right)$

Rendering Operation $\mathcal{R}$ →

(a) Scene Language     (b) Output Entity     (c) Reparametrized Entity    (d) Rendered Image

Figure 3: **Rendering.** Given a Scene Language in (a), a program interpreter executes the program to obtain a data object in (b). A graphics renderer first reparameterizes the data object from (b) into the renderer-specific parameter space, and then executes the rendering operation $\mathcal{R}$ to obtain final image outputs in (d).

| Renderer | Examples | | |
|---|---|---|---|
| | Rendering Operation $\mathcal{R}$ | Parameters from $\Theta$ | $g_{\text{reparam}}$ |
| Primitive-based renderer | Light transport simulation | Shape and BRDF parameters | LM inference |
| Asset-based renderer | Ray tracing | Asset metadata | LM inference |
| SDS-based renderer | Gaussian splatting | 3D Gaussian parameters | SDS optimization |
| T2I model | Model feed-forward pass | Text embeddings in $\mathcal{Z}$ | CLIP text encoding |

Table 3: **Examples of Graphics Renderers.** The module specification for graphics renderer from Fig. 3 can be instantiated with different rendering approaches.

The data type of an entity $h = f_w(z)$ is denoted as `Entity`, which is recursively defined as a nested list. At the base level, an entity data object has three data fields of types `Word`, `Embedding`, and `Matrix`. These three fields describe the entity's semantic group, identity, and pose in the frame of $h$, respectively. In particular, `Embedding` captures the visual details of entities and requires a highly expressive representation, such as neural embeddings. In this work, we employ the textual embedding space of OpenCLIP-ViT/H (Ilharco et al., 2021) for attribute parameterization, denoted as $\mathcal{Z}_{\text{CLIP}}$. It offers the advantage that embeddings can be either encoded directly from natural language descriptions or inferred from images with Textual Inversion (Gal et al., 2023). Table 1 summarizes the operations and data types in accordance with the notations introduced in Section 3.1.

## 4 RENDERING

Applying the proposed scene representation to image generation tasks requires rendering a Scene Language $\Phi(s)$ into images. To do so, first, the program interpreter evaluates $\Phi(s)$ to obtain a data object of type `Entity`. Afterward, a graphics renderer maps the `Entity` data object to its rendering parameter space and renders it into a final image.

**Renderer Specifications.** We define the specification of a graphics renderer, a module in the proposed representation, as follows. A graphics renderer is determined by (1) primitive parameter space $\Theta$ and (2) a rendering operation $\mathcal{R} : \mathcal{P}(\Theta \times \mathcal{T}) \to \mathcal{I}$, where $\mathcal{T}$ is the space of 3D affine transformations representing poses, $\mathcal{P}$ denotes all possible subsets, and $\mathcal{I}$ is the space of rendered images. We assume access to a reparameterization function $g_{\text{reparam}}$ that maps from `Tuple[Word, Embedding]` to $\Theta$, which consequently determines a mapping from program outputs of type `Entity` to the admissible input domain of rendering operation $\mathcal{R}$.

**Renderer Instantiations.** An example renderer instantiation is with Score Distillation Sampling (SDS) (Poole et al., 2022) guidance, where $\Theta$ is a differentiable 3D representation and $g_{\text{reparam}} : \mathcal{Z}_{\text{CLIP}} \to \Theta$ corresponds to the SDS-guided optimization process of finding a solution that aligns with the input of $g_{\text{reparam}}$. To compute $z$ given a word, *e.g.*, "pawn" for an entity of white pawn from Fig. 2, and an embedding, *e.g.*, `<z68>`, we use a manually specified language template $c$, or "a pawn, `<z68>`, 3D model" in this example, to embed them into embedding $z = g_{\text{CLIP}}(c) \in \mathcal{Z}_{\text{CLIP}}$; $g_{\text{CLIP}}$ is the pre-trained CLIP text encoder.

For the underlying 3D representation, we use 3D Gaussian Splatting (Kerbl et al., 2023) where images are rendered by splatting a set of 3D Gaussians onto the image plane; other differentiable 3D representations such as neural fields will also be suitable. We base our implementation on GALA3D (Zhou et al., 2024c), and use MVDream (Shi et al., 2023) and a depth-conditioned ControlNet (Zhang et al., 2023b) for guidance.

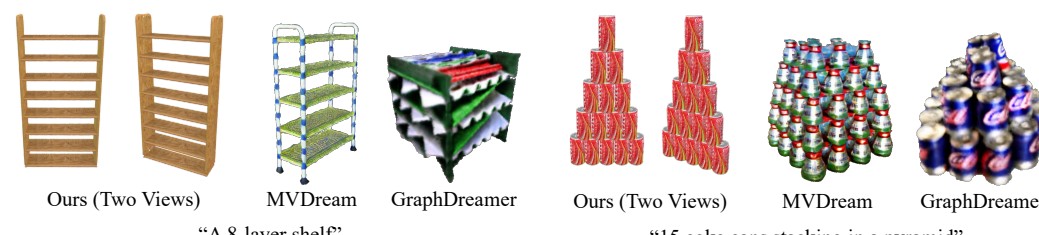

Ours (Two Views)    MVDream    GraphDreamer      Ours (Two Views)    MVDream    GraphDreamer

"A 8-layer shelf"          "15 coke cans stacking in a pyramid"

Figure 4: **Text-Conditioned Scene Generation.** Input text prompts are shown at the bottom of each row. Compared to using no intermediate representation (MVDream) or scene graph (GraphDreamer), our Scene Language results in more detailed and accurate outputs.

We will refer to the renderer above as the Gaussians renderer. Other possible renderers include primitive-based renderers, such as Mitsuba (Jakob et al., 2022) with graphics primitives of cubes, spheres, and cylinders, asset-based game engines, such as MineCraft[2], and feed-forward inference of layout-conditioned text-to-image (T2I) diffusion models, such as MIGC (Zhou et al., 2024a), which achieves 2D bounding box conditioning by controlling attention layers from Stable Diffusion (Rombach et al., 2022)). A summary is shown in Table 3 and details are deferred to Appendix D.

## 5   INFERENCE VIA PRE-TRAINED LANGUAGE MODELS

We introduce a training-free method to infer a Scene Language from text or image descriptions of scenes using pre-trained language models (LMs). LMs have shown remarkable capability in code generation with common programming languages such as Python. In our implementation, we prompt LMs to generate a Python program, which is further executed with a program interpreter and rendered into an image using a graphics renderer.

In particular, we include the following in the LM prompt: 1) the input condition, which is a scene description in texts or an image; 2) a Python script of helper functions converted from the macros from the DSL; and 3) an example program using the helper functions. We use Claude 3.5 Sonnet (Anthropic, 2024) for all experiments for our method and LM-dependent baselines. Full language prompts for all experiments are listed in Appendix E.

Recall from Section 3.1 that functions in program $P$ are evaluated on embeddings from $Z$. The function arguments in the LM-generated programs, which are numeric values or string tokens, are converted to embeddings from $\mathcal{Z}_{\text{CLIP}}$ (Section 3.2) using language templates and the CLIP text encoder $g_{\text{CLIP}}$. For example, in the LM-generated program, function calls for white pieces in Fig. 2 have input attribute $\{$``color'':(.9,.9,.9)$\}$, and we prompt LM to describe the color value as a word, and feed the word into $g_{\text{CLIP}}$ to compute `<z68>`. For image-conditioned tasks, for each primitive entity in the execution output of $P$, we first use GroundingSAM (Kirillov et al., 2023; Ren et al., 2024) to segment out the region defined by the `word` associated with the entity. We then use Textual Inversion (Gal et al., 2023) to optimize an embedding to reconstruct the cropped image with the diffusion model training objective. The full process is deferred to Appendix F.1.

## 6   APPLICATIONS

We apply the inference method from Section 5 to the tasks of text-conditioned 3D scene generation (Section 6.1) and editing (Section 6.2), image-conditioned scene generation (Section 6.3), and 4D scene generation (Section 6.4).

### 6.1   TEXT-CONDITIONED SCENE GENERATION

This task aims to synthesize scenes conditioned on a textual scene description.

**Baselines.** To evaluate the proposed representation, we compare our inference pipeline with 3D scene generation methods using alternative intermediate representations, *e.g.*, scene graph. In particular, we compare with GraphDreamer (Gao et al., 2024) as an exemplar approach, which generates scene graphs from input texts via LM prompting and then synthesizes scenes conditioned on the graphs via SDS guidance. We further ablate the role of structural representation in this task by comparing ours with the backbone of our SDS-based renderer, MVDream (Shi et al., 2023), as a direct scene generation approach. Full implementation details in Appendix F.2.

---

[2]https://www.minecraft.net

**Results.** Text-conditioned scene generation results rendered with the SDS-based renderer are shown in Fig. 4. Compared to the direct 3D scene generation method MV-Dream, our approach is compositional and adheres more closely to input prompts in scenes involving multiple objects. Compared to a scene graph representation, where entity relations are restricted to be between two objects and are bottlenecked by the coarseness of natural language descriptions, *e.g.*, "aligned in a row", a program-based representation offers more flexible and precise specifications for relations, *e.g.*, the particular coke can arrangement in Fig. 4. This brings the practical benefit of offloading the burden of generating scenes involving complex entity relations from the T2I model (used for SDS guidance in both ours and GraphDreamer) towards LM, leading to accurate and detailed generation results.

| Methods | Alignment | Counting |
|---|---|---|
| GraphDreamer | $3.56_{\pm 7.38}$ | 0.11 |
| MVDream | $10.79_{\pm 12.83}$ | 0.11 |
| Ours | $\mathbf{85.65_{\pm 13.71}}$ | **1.0** |

Table 4: **Quantitative Evaluation Results**. We perform a user study to compare with prior methods on the text-conditioned 3D generation task and report the percentages of user preferences for prompt alignment. We also report the counting accuracy (0 for inaccurate and 1 for accurate). Results are averaged across 9 scene prompts and 103 users; ± denotes standard deviation.

To quantitatively compare our method with baselines, we conduct a user study on Prolific[3] and ask users to choose one of the three animated scenes, synthesized by ours and two baselines in a randomized order, that aligns the best with the text prompt for the scene. Details are deferred to Appendix F.3. We further report whether the synthesized scenes have the correct object count. As shown in Table 4, our method achieves a more favorable prompt alignment than the baselines and has a clear advantage in counting accuracy.

## 6.2 TEXT-INSTRUCTED SCENE EDITING

Scenes synthesized from our proposed representation can further be edited following natural language instructions by prompting LM with its previously generated program and an editing instruction. The results are shown in Fig. 5. Our representation provides an interpretable and intuitive interface for scene editing, as 1) functions have explicit semantic meanings associated with words, and 2) function reuse greatly improves the readability of programs. Furthermore, since the structure of programs reflects the structure of scenes, editing program parameters leads to changes in the scenes while preserving the original structure, *e.g.*, the circular arrangement of staircases in Fig. 5. The desirable editing effects involving multiple primitives, or all staircases in this example, can be effectively achieved via only small changes in the program space. Finally, the program structure itself, *e.g.*, the function header in the Jenga set example, can be adjusted for editing, achieving localized edits that only affect relevant parts of the scene.

The composibility of our representation directly benefits localized scene editing. In comparison, MVDream from Section 6.1 does not apply to this task, as the full scene is parameterized with a single 3D representation. Precisely encoding the geometric relations of scene components further enhances the controllability of generated scenes. In comparison, GraphDreamer represents the binary relation of scene components with coarse language descriptions and therefore does not apply to editing tasks involving precise geometric controls, *e.g.*, in the first example from Fig. 5.

## 6.3 IMAGE-CONDITIONED SCENE GENERATION

We further show that the proposed representation can be used for image parsing and generating 3D scenes consistent with the parsed image structure and content.

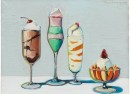 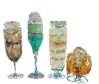 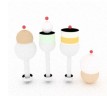 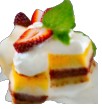

Input Image    Ours (Gaussians)    Ours (Mitsuba)    GraphDreamer

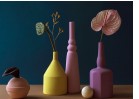 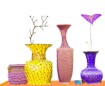 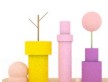 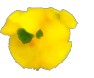

Input Image    Ours (Gaussians)    Ours (Mitsuba)    GraphDreamer

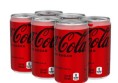 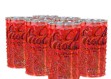 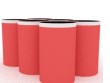 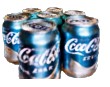

Input Image    Ours (Gaussians)    Ours (Mitsuba)    GraphDreamer

Figure 6: **Image-Conditioned Scene Generation.** Both our method and GraphDreamer parse an input image to semantic entities. Compared to the baseline, programs from our representation encode additional scene structure, *e.g.*, repetitions, and specify geometric relations among entities more precisely. Embeddings from ours further enable visual identity preservation in the renderings.

---

[3]https://www.prolific.com/

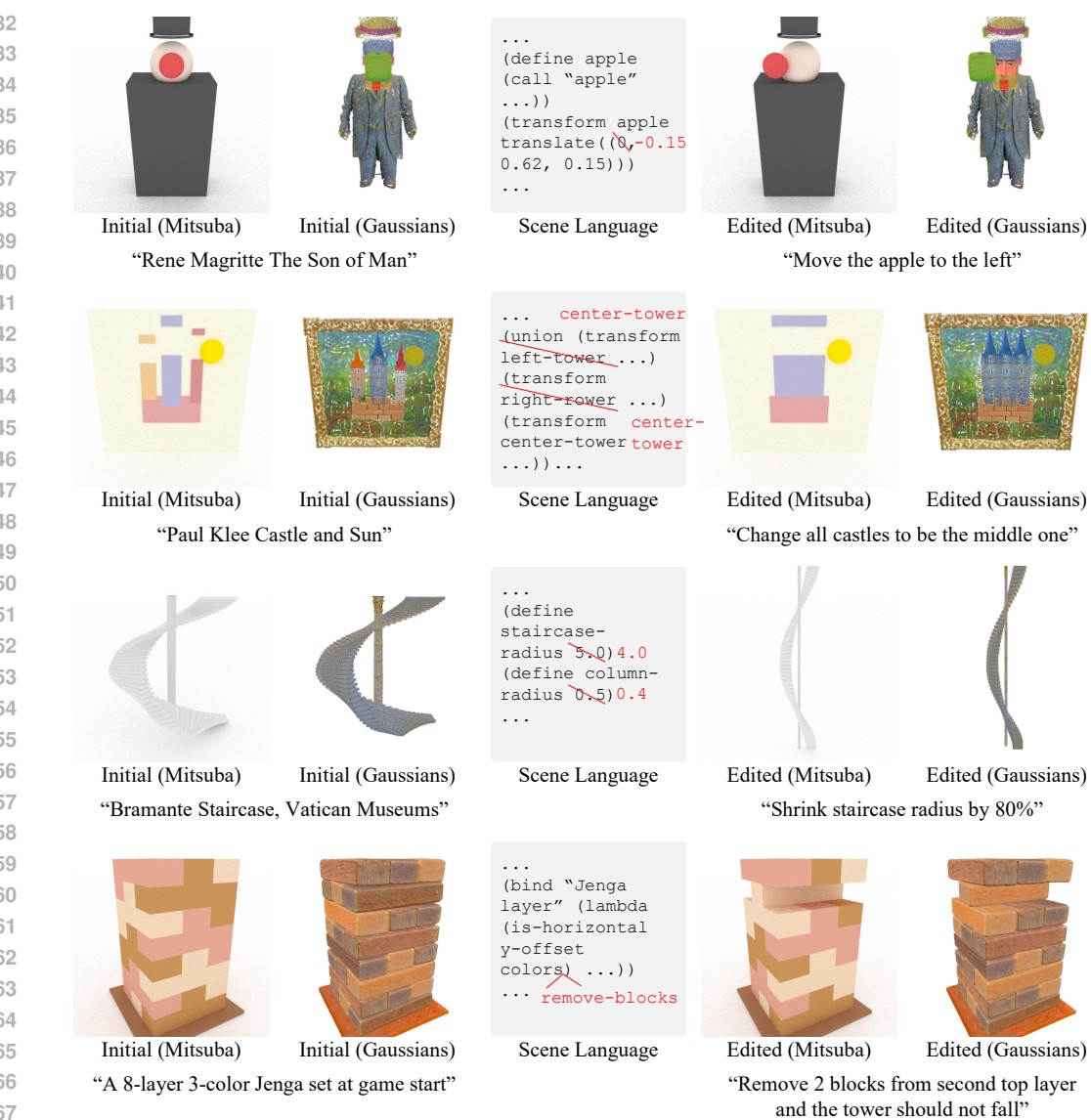

Figure 5: **Scene Editing with Language Instructions.** The program structure from our representation is highly interpretable, which benefits user editing. The bottom of each row shows initial scene descriptions and editing instructions in the format of user text prompts. We prompt an LM to infer the initial Scene Language as well as the edits (shown with texts in red), and show image renderings with two renderers.

We compare our representation with scene graphs by comparing our method with GraphDreamer. The results are shown in Fig. 6. The proposed representation explicitly encodes 1) semantic components parsed from input images, 2) the high-level scene structures, *e.g.*, the repetition of coke cans, and 3) visual details, *e.g.*, the glass bottles with particular shapes and colors. Compared with our method, which preserves both structure and visual content from input images, GraphDreamer only reconstructs semantics from input images and leaves out entity poses and identities, due to the information loss in the intermediate scene graph representation.

### 6.4 TEXT-CONDITIONED 4D SCENE GENERATION

We apply the inference method from Section 5 to generate 4D scenes. The 4D scene representation in this task is identical to the definition in Eq. (1), except that there is an additional 4D entity function in the program $P$. The corresponding DSL extends from Table 2 as specified in Appendix C.

Allowing for a flexible set of primitive entities is a crucial property of our representation that makes it suitable for generating diverse 4D scenes of different scales, including objects with moving parts (*e.g.*, the wind turbine from Fig. 7) and scenes with moving objects (*e.g.*, the carousel). Specifically,

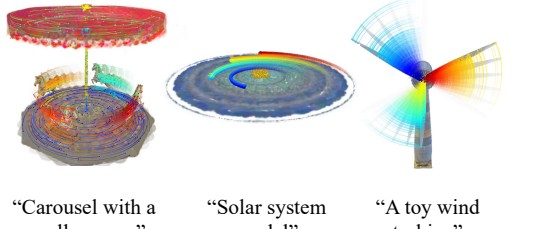

"Carousel with a small canopy"    "Solar system model"    "A toy wind turbine"



(a) Rendering (First Frame)    (b) Semantic Segmentation    (c) Instance Segmentation    (d) Instance Correspondence

Figure 7: **Text-Conditioned 4D Scene Generation.** The proposed representation captures the structure not only for static, but also for dynamic scenes, and can be applied for synthesizing 4D scenes. It explicitly represents the temporal correspondence of an entity in a dynamic scene. Each colored trajectory denotes tracking of a temporally moving point.

Figure 8: **Visualizations of Discriminative Maps.** The proposed representation contains semantics information for scene components, visualized using semantic segmentation shown in (b). It is compositional and directly informs instance segmentation (c). Furthermore, it specifies the dense correspondence across repeated entities (d).

primitives have granularity adapted to the particular scene being represented, instead of being chosen from a fixed set (Tian et al., 2019; Sharma et al., 2018) or object-centric as in scene graphs (Johnson et al., 2015).

Moreover, the hierarchical scene structure encapsulated by our program-based representation makes it possible to represent 4D scenes compactly, serving as a regularization for generation output. Entities (*e.g.*, multiple horses from the function "horse" from the carousel scene in Fig. 7) can be grouped into one function ("horses") and thereby share the same temporal transformation. Writing composible functions for entity grouping effectively reduces the dimension of the temporal motion space and improves motion fidelity. See Appendix B for better visualizations.

### 6.5 DIFFERENT GRAPHICS RENDERERS

The same program can be rendered with different renderers described in Section 4, showing the versatility of the proposed representation. The results are shown in Fig. 9 with the same experiment setup as in Section 6.1.

### 6.6 VISUALIZATION OF DISCRIMINATIVE INFORMATION

As shown in Fig. 8, several pieces of discriminative information can be directly obtained with the proposed representation: semantic maps in (b), as words represent per-entity semantics; instance segmentation in (c), as the representation is compositional with separable instances; correspondence of the repeated instances in (d), as programs specify repetitions existing in a scene; dense temporal correspondence for 4D scenes, as shown in Fig. 7.

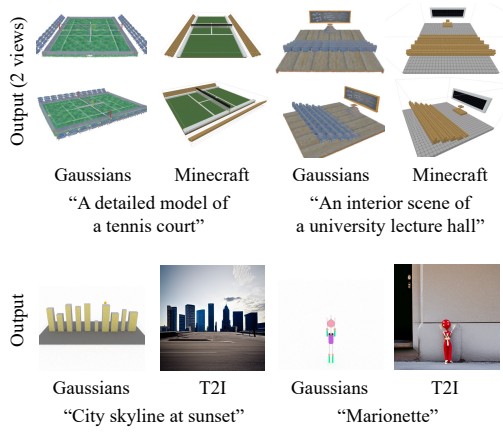

Output (2 views)

Gaussians    Minecraft    Gaussians    Minecraft

"A detailed model of a tennis court"    "An interior scene of a university lecture hall"

Output

Gaussians    T2I    Gaussians    T2I

"City skyline at sunset"    "Marionette"

Figure 9: **Renderings Across Graphics Renderers.** Different renderers produce renderings that adhere to the same representation and therefore are visually aligned, while each exhibits a different imaging style. Text inputs are shown at the bottom of the subfigures.

## 7 CONCLUSION

We have introduced a visual scene representation, termed the Scene Language, which encodes three key aspects of visual scenes: (1) scene structure, such as hierarchy and repetition, specified via programs; (2) semantics of individual scene components succinctly summarized via words in natural language; and (3) identities of each component precisely captured via neural embeddings. We formalize the representation as a programming language defined using a DSL. We show that the proposed representation can be efficiently inferred from both text and image inputs using pre-trained language models. Once the program is executed, the resulting scene can be rendered into images using a variety of graphics renderers. Compared with existing methods, our Scene Language produces 3D and 4D scenes with significantly higher fidelity, preserves complex scene structures, and enables easy and precise editing.

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

| Macros | | |
|---|---|---|
| translate | :: | Vector -> Matrix   // Compute translation matrix |
| rotate | :: | Float -> Vector -> Vector -> Matrix   // Compute rotation matrix |
| scale | :: | Vector -> Vector -> Matrix   // Compute scaling matrix |
| reflect | :: | Vector -> Vector -> Matrix   // Compute reflection matrix |
| @ | :: | Matrix -> Matrix -> Matrix   // Matrix multiplication |
| compute-shape-center | :: | Entity -> Vector   // Compute center of an entity's bounding box |
| compute-shape-min | :: | Entity -> Vector   // Compute minimum corner of an entity's bounding box |
| compute-shape-max | :: | Entity -> Vector   // Compute maximum corner of an entity's bounding box |
| compute-shape-sizes | :: | Entity -> Vector   // Compute sizes of an entity's bounding box |

Table 5: **The Domain-Specific Language** includes the definitions from Table 2 and the transformation-related macros from this table.

| Grammar | |
|---|---|
| 4D-entity-func | ::= (lambda () create-entity-list)   // Define a function that outputs a 4D scene |
| create-entity-list | ::= (list create-entity*)   // Represent a 4D scene as a temporal list of entities |

Table 6: **The Domain-Specific Language** for 4D scenes. * indicates one or more expressions.

## A  OVERVIEW

The supplementary contains the following content: additional qualitative results (Appendix B), followed by details for the representation definition (Appendix C), graphics renderers (Appendix D), and experiments (Appendix F). Please refer to main text to see how they are integrated.

## B  ADDITIONAL RESULTS

Please refer to the webpage `https://sclg-page.github.io/` for animated results.

## C  DOMAIN-SPECIFIC LANGUAGE

The complete DSL includes the ones listed in Table 2, with additional macros for computing transformation matrices as listed in Table 5, and grammar for 4D scenes as listed in Table 6.

## D  DETAILS OF GRAPHICS RENDERERS

This section expands the instantiation of three graphics renderers from Section 4 in detail. For each rendere, we will discuss its parameter space $\Theta$ and $\mathcal{T}$, renderer $\mathcal{R}$, and the reparameterization function $g_{\text{reparam}}$.

### D.1  SDS-BASED RENDERER

**Parameter Space with 3D Gaussians.** For this renderer, $\Theta$ is the space of 3D Gaussian parameters and $\mathcal{T}$ is the space of 3D affine transformation matrices. In particular, each primitive is parameterized as a set of $K$ 3D Gaussians under a 3D affine transformation $t$, written as $(\theta, t) = (K, \{\phi_i\}_{i=1}^{K}, t) \in \Theta \times \mathcal{T}$, where $\phi_i$ is the set of parameters for a single 3D Gaussian, and $t$ is a 3D transformation matrix. Each Gaussian parameter $\phi$ is defined as $\phi := (\mu, \alpha, s, q, c)$, denoting the 3D center position, opacity, scale, rotation in quaternion, and color of the Gaussian, respectively. An entity consisting of $N$ primitives is parameterized as $\{(\theta_j, t_j)\}_{j=1}^{N} = \{(K_j, \{\phi_i^j\}_{i=1}^{K_j}, t_j)\}_{j=1}^{N}$.

**Differentiable Rendering.** The rendering operation $\mathcal{R}$ for the 3D Gaussian renderer is as follows.

Following Kerbl et al. (2023), a single Gaussian is defined by

$$G(x) = e^{-\frac{1}{2}(x-\mu)^T \Sigma^{-1}(x-\mu)},$$

where $x \in \mathbb{R}^3$ is a point in world coordinate, $\Sigma := (RS)(RS)^T$ the 3D covariance matrix, $R$ the rotation matrix computed from $q$, and $S$ the scaling matrix computed from $s$.

A Gaussian under transformation $t \in \mathcal{T}$ with $t(x) = R_t S_t x + p_t$, where $R_t, S_t, p_t$ are the rotation, scaling, and translation components, respectively, is then computed with $G_t$ satisfying the follows:

$$G_t(t(x)) = G(x).$$

We assume that diagonal entries of the scaling matrix $S_t$ are all positive, and therefore $t$ is invertible. Combining the above gives

$$G_t(x) = e^{-\frac{1}{2}(x-\mu_t)^T \Sigma_t^{-1}(x-\mu_t)},$$

where $\mu_t = t(\mu)$ and $\Sigma_t = ((R_t R)(S_t S))((R_t R)(S_t S))^T$. Let $\tilde{t}(\phi)$ be the Gaussian after applying transformation $t$ on $\phi$. Then $\tilde{t}(\phi)$ has center $\mu_t$, rotation $R_t R$, scale $S_t S$, and has $\alpha$ and $c$ remaining unchanged as derived above.

The rendering operation $\mathcal{R}$ to convert an entity consisting of $N$ primitives, $\{(\theta_j, t_j)\}_{j=1}^N = \{(K_j, \{\phi_i^j\}_{i=1}^{K_j}, t_j)\}_{j=1}^N$, to the image space simply amounts to rendering all post-transformation 3D Gaussians in the scene, $\{\tilde{t}_j(\theta_j)\}_j := \{\tilde{t}_j(\phi_i)\}_{i,j}$, following the projection and blending process from Kerbl et al. (2023).

**Primitive Reparameterization via SDS Guidance.** Recall that $g_{\text{reparam}}$ aims to obtain 3D Gaussian primitive parameters for per-primitive conditional embeddings $\{z_j\}_{j=1}^N$ and global condition $z_{\text{global}}$, where $z_j = g_{\text{CLIP}}(c_j)$ is explained in Section 4, and $z_{\text{global}} = g_{\text{CLIP}}(c_{\text{global}})$ is computed from a global scene description in texts, $c_{\text{global}}$. We now expand Section 4 to describe the optimization process of $g_{\text{reparam}}$ in detail.

We write the SDS objective originally proposed in Poole et al. (2022) as follows:

$$g(\psi; z, \hat{\epsilon}) := \nabla_\psi \mathcal{L}_{\text{SDS}}(x = \mathcal{R}(\psi); z, \hat{\epsilon}) = \mathbb{E}_{\eta \sim \mathcal{U}(0,1), \epsilon \sim \mathcal{N}(0,I)} \left[ w(\eta)(\hat{\epsilon}(\alpha_\eta x + \alpha_\eta \epsilon, z, \eta) - \epsilon) \frac{\partial x}{\partial \psi} \right],$$

where $\hat{\epsilon}$ is a pre-trained image denoising network, $\eta$ is diffusion timestep, and $w(\cdot), \alpha_\eta$ come from diffusion schedule.

With the notations from above, for entity $\{(\theta_j, t_j)\}_{j=1}^N$, let

$$\mathcal{L}(\{z_j\}_j, z_{\text{global}}, \{t_{\text{init},j}\}_j) := \mathcal{L}_{\text{SDS}}(\{\tilde{t}_j(\theta_j)\}_j; z_{\text{global}}, \hat{\epsilon}_{\text{ControlNet}}) + \sum_j \mathcal{L}_{\text{SDS}}(\theta_j; z_j, \hat{\epsilon}_{\text{MVDream}})$$

$$+ \sum_j \mathcal{L}_{\text{reg}}(\theta_j, \text{StopGrad}(t_j)) + \sum_j \mathcal{L}_{\text{layout}}(\theta_j, t_{\text{init},j}),$$

where $\mathcal{L}_{\text{reg}}, \mathcal{L}_{\text{layout}}$ are regularization terms following the definition from Zhou et al. (2024c) and StopGrad stops gradients from backpropagation. Here, $\mathcal{L}_{\text{reg}}$ penalizes Gaussian ellipsoids that are too long, and $\mathcal{L}_{\text{layout}}$ penalizes Gaussians that lie outside the intial bounding box specified by $t_{\text{init}}$.

Finally, we have

$$g_{\text{reparam}} = \arg \min_{\{(\theta_j, t_j)\}_{j=1}^N} \mathcal{L}.$$

During optimization, if primitives $j_1$ and $j_2$ have the same condition and initial normalized bounding box scale, *i.e.*, $(z_{j_1} = z_{j_2}) \wedge (\frac{S_{t_{j_1}}}{\|S_{t_{j_1}}\|_2} = \frac{S_{t_{j_2}}}{\|S_{t_{j_2}}\|_2})$, they are enforced to have the same parameters $\theta$ (but still distinct $t_{j_1}$ and $t_{j_2}$), which greatly reduces the number of parameters in the solution space.

In practice, for certain scenes, LM outputs treat detailed object parts as primitives, *e.g.*, the hat rim and hat top from the first example in Fig. 5, and the backbone model for SDS guidance cannot effectively model such fine-grained parts. Therefore, we treat the hat as a primitive, whose pose is computed from the minimum bounding box containing both the hat rim and hat top, before carrying out the above optimization. This process effectively adapts the granularity of the computation graph, originally specified in LM inference outputs, to the graphics renderer being used, by assigning intermediate nodes from the original computation graph as the new leaf nodes.

### D.2 MITSUBA RENDERER

**Parameter Space.** For this renderer, $\Theta$ is the parameter space for three types of graphics primitives supported by Mitsuba: `cube`, `sphere`, and `cylinder`, as specified in the function header for `primitive_call` in Appendix E.1. $\mathcal{T}$ is the 3D affine transformation space.

**Renderer.** We use the path tracer with maximum depth 8 implemented in Mitsuba.

**Reparameterization.** Since we directly prompt LM to generate Mitsuba primitive parameters in its outputs as specified in Appendix E.1, the function parameters from raw LM outputs are already in the parameter space $\Theta$ and are directly used for rendering, instead of being encoded into CLIP embeddings $z \in \mathcal{Z}_{\text{CLIP}}$.

### D.3 MINECRAFT RENDERER

**Parameter Space.** For this renderer, $\Theta$ is the asset parameters for Mincraft blocks, and $\mathcal{T}$ is the space of 3D similarity transformation matrices, *i.e.*, of scaling and translation transformations. Note that we prevent rotation transformations in Minecraft, since that could lead to shapes that are impossible to render correctly in Minecraft.

Specifically, $\Theta$ is specified in the docstring from Appendix E.4 and is expanded below. We introduce two types of primitives that let us construct in-game elements.

The first is `set_cuboid`. This primitive facilitates the creation of a cuboid within the Minecraft environment. The function accepts three arguments: (1) A string denoting the Minecraft block type (*e.g.*, `minecraft:white_concrete`); (2) A tuple of three integers representing the scaling along the x, y, and z axes; (3) A boolean flag, `fill`, that specifies whether the cuboid should be solid or hollow. The cuboid is anchored at the coordinate origin $(0, 0, 0)$, which corresponds to its front-left-bottom vertex.

The second is `delete_blocks`. This primitive allows for the deletion of a previously placed cuboid. It accepts a single parameter, which is a tuple of three integers denoting the scaling along the x, y, and z axes. This operation removes the cuboid with its front-left-bottom vertex at the origin $(0, 0, 0)$, effectively clearing the designated space.

Note that we do not provide the Minecraft block type in the prompt, but instead let the model choose this parameter. Since there is a large amount of Minecraft data files on the web, the model performs decently well in choosing appropriate Minecraft blocks. We also augment this by building safety checks; for example, if the model chooses a Minecraft block that doesn't exist in our version of Minecraft, we will use semantic similarity to choose the most similar block from our library.

We also are able to translate easily from Minecraft renderings to Mitsuba renderings, by converting Minecraft blocks to corresponding cuboids in Mitsuba. We also color the Mitsuba blocks accordingly to the average color of the Minecraft block.

**Renderer.** We use WebGL[4] and Deepslate[5] for rendering Minecraft builds.

**Reparameterization.** Similar to Mitsuba, function parameters from LM-generated programs are directly used for rendering without CLIP encoding or parameterization.

### D.4 TEXT-TO-IMAGE (T2I) MODEL RENDERER

**Parameter Space.** We employ MIGC (Zhou et al., 2024a) as the backbone model for this renderer, which originally uses a CLIP text encoder (Radford et al., 2021) and a pre-trained UNet from Stable Diffusion (Rombach et al., 2022) for layout-conditioned text-to-image generation. The parameter space $\Theta$ for this renderer is the CLIP text embedding space.

**Renderer.** We first project the 3D bounding boxes of primitives from an execution output of our representation to a 2D layout under a specified camera viewpoint, and then run the forward pass of the T2I model conditioned on the 2D layout, where each 2D bounding box corresponds to an aforementioned CLIP embedding $\theta \in \Theta$ .

**Reparameterization.** Function parameters from LM-generated programs are directly encoded by the CLIP text encoder using the language templates described in Section 5.

---

[4]https://get.webgl.org/

[5]https://misode.github.io/deepslate/

## E LANGUAGE MODEL PROMPTS

### E.1 TEXT- AND IMAGE-CONDITIONED SCENE GENERATION

In Section 5, we introduced an inference method for the representation by prompting LMs. The full system prompt is displayed below. The system prompt defines the data types and the function headers of macros from the DSL, written in Python.

```python
You are a code completion model and can only write python functions wrapped within
python.

You are provided with the following helper.py which defines the given functions and
definitions:

"""This module contains a Domain-Specific Language (DSL) designed
with built-in support for loops and functions for shape construction and transformation.
"""

from typing import NamedTuple, Any, Callable, Literal
import math
import numpy as np

# type aliases and DSL syntax sugar
P = Any  # 3D vector, e.g., a point or direction
T = Any  # 4x4 transformation matrix
Shape = list[dict[str, Any]]  # a shape is a list of primitive shapes

# shape function library utils

def register(docstring: str):
    """
    Registers a function whose name must be unique. Provide keyword argument defaults for
    ↪ easier debugging.
    """
def library_call(func_name: str, **kwargs) -> Shape:
    """
    Call a function from the library and return its outputs. You are responsible for
    ↪ registering the function with `register`.

    Args:
        func_name (str): Function name.
        **kwargs: Keyword arguments passed to the function.
    """

def primitive_call(name: Literal['cube', 'sphere', 'cylinder'], shape_kwargs: dict[str,
    ↪ Any], color: tuple[float, float, float] = (1.0, 1.0, 1.0)) -> Shape:
    """
    Constructs a primitive shape.

    Args:
        name: str - 'cube', 'sphere', or 'cylinder'.
        shape_kwargs: dict[str, Any] - keyword arguments for the primitive shape.
            - For 'cube': {'scale': P} - 3-tuple of floats for scaling along x, y, z
                ↪ axes.
            - For 'sphere': {'radius': float} - radius of the sphere.
            - For 'cylinder': {'radius': float, 'p0': P, 'p1': P}
                - radius: float - radius of the cylinder.
                - p0: P - 3-tuple of floats for the start point of the cylinder's
                    ↪ centerline.
                - p1: P - 3-tuple of floats for the end point of the cylinder's
                    ↪ centerline.
        color: Tuple[float, float, float] - RGB color in range [0, 1]^3.

    Returns:
        Shape - the primitive shape.

    Examples:
        - `primitive_call('cube', shape_kwargs={'scale': (1, 2, 1)})`
          Returns a cube with corners (-0.5, -1, -0.5) and (0.5, 1, 0.5).
        - `primitive_call('sphere', shape_kwargs={'radius': 0.5})`
          Returns a sphere with radius 0.5, with bounding box corners (-0.5, -0.5, -0.5)
          ↪ and (0.5, 0.5, 0.5).
        - `primitive_call('cylinder', shape_kwargs={'radius': 0.5, 'height': 1})`
          Returns a cylinder with radius 0.5, height 1, with bounding box corners (-0.5,
          ↪ -0.5, -0.5) and (0.5, 0.5, 0.5).
    """

# control flows

def loop(n: int, fn: Callable[[int], Shape]) -> Shape:
    """
    Simple loop executing a function `n` times and concatenating the results.

    Args:
```

```
        n (int): Number of iterations.
        fn (Callable[[int], Shape]): Function that takes the current iteration index
        ↪   returns a shape.

    Returns:
        Concatenated shapes from each iteration.
    """

# shape manipulation

def concat_shapes(*shapes: Shape) -> Shape:
    """
    Combines multiple shapes into a single shape.
    """
def transform_shape(shape: Shape, pose: T) -> Shape:
    """
    Args:
        shape: Shape
        pose: T - If pose is A @ B, then B is applied first, followed by A.

    Returns:
        The input shape transformed by the given pose.
    """

# pose transformation

def rotation_matrix(angle: float, direction: P, point: P) -> T:
    """
    Args:
        angle (float) : the angle of rotation in radians
        direction (P) : the axis of rotation
        point (P) : the point about which the rotation is performed
    """
def translation_matrix(offset: P) -> T:
    """
    Args:
        offset (P) : the translation vector
    """
def scale_matrix(scale: float, origin: P) -> T:
    """
    Args:
        scale (float) - the scaling factor, only uniform scaling is supported
        origin (P) - the origin of the scaling operation
    """
def identity_matrix() -> T:
    """
    Returns the identity matrix in SE(3).
    """

# calculate locations and sizes of shape bounding boxes

def compute_shape_center(shape: Shape) -> P:
    """
    Returns the shape center.
    """
def compute_shape_min(shape: Shape) -> P:
    """
    Returns the min corner of the shape.
    """
def compute_shape_max(shape: Shape) -> P:
    """
    Returns the max corner of the shape.
    """
def compute_shape_sizes(shape: Shape) -> P:
    """
    Returns the shape sizes along x, y, and z axes.
    """

STRICTLY follow these rules:

    1. Only use the functions and imported libraries in helper.py.

    2. You can only write functions.  Follow a modular approach and use the register
       decorator to define semantic shapes or shape groups.

    3. Camera coordinate system:  +x is right, +y is up, +z is backward.

    4. You can use shape primitives to approximate shape components that are too
       complex.  You must make sure shape have correct poses.  Be careful about set_mode
       and set_to from primitive_call.

    5. You must use library_call to call registered functions.

    6. Use compute_shape_* from helper.py if possible to compute transformations.

You should be precise and creative.
```

The full user prompt for image or text-conditioned 3D generation is displayed below. It includes an example valid program, and the task specification indicated with a placeholder {task}. For text-conditioned generation (Section 6.1), it is replaced with the input textual scene description. For image-conditioned generation (Section 6.3), it is replaced with ``Reconstruct the input scene'', and the input image is also fed into LM.

```python
Here are some examples of how to use helper.py:

from helper import *

"""
A pile of books on a desk
"""

@register("book")
def book(scale: P) -> Shape:
    return primitive_call('cube', color=(.6, .3, .1), shape_kwargs={'scale': scale})

@register("books")
def books(width: float, length: float, book_height: float, num_books: int) -> Shape:
    def loop_fn(i) -> Shape:
        book_shape = library_call('book', scale=(width, book_height, length))
        book_shape = transform_shape(book_shape,
        ↪   translation_matrix([np.random.uniform(-0.05, 0.05), i * book_height,
        ↪   np.random.uniform(-0.05, 0.05)]))  # FIRST translate
        book_center = compute_shape_center(book_shape)  # must be computed AFTER
        ↪   transformation!!
        return transform_shape(book_shape, rotation_matrix(np.random.uniform(-0.1, 0.1),
        ↪   direction=(0, 1, 0), point=book_center))  # THEN tilt

    return loop(num_books, loop_fn)

@register("desk")
def desk(scale: P) -> Shape:
    return primitive_call('cube', color=(.4, .2, .1), shape_kwargs={'scale': scale})

@register('desk with books')
def desk_with_books() -> Shape:
    desk_shape = library_call('desk', scale=(1, .1, .5))
    books_shape = library_call('books', width=.21, length=.29, book_height=.05,
    ↪   num_books=3)
    _, desk_top, _ = compute_shape_max(desk_shape)
    _, books_bottom, _ = compute_shape_min(books_shape)
    return concat_shapes(
        desk_shape,
        transform_shape(books_shape, translation_matrix((0, desk_top - books_bottom, 0)))
        ↪   # stack books on top of desk
    )

IMPORTANT: THE FUNCTIONS ABOVE ARE JUST EXAMPLES, YOU CANNOT USE THEM IN YOUR PROGRAM!

Now, write a similar program for the given task:

from helper import *

"""
{task}
"""
```

## E.2   SCENE EDITING

For scene editing (Section 6.2), we prompt the LM in two rounds, first with a textual scene description with the same protocol from Section 6.1, and then with an editing instruction, *e.g.*, ``move the apple to the left''. In the second round, the system prompt remains the same as Appendix E.1. The user prompt is as follows, where {program} is the LM output from first round, and {task} is the editing instruction.

```
Here is a program using helper.py:

{program}

Now, do minimal edit to the program such that the scene function, when called, will
follow the instruction: {task}.  Your code starts here.
```

```
1026    from helper import *
1027
1028    """
        {task}
1029    """
```

1030

1031

### E.3   4D GENERATION

For 4D generation, we include one more macro definition in the system prompt as shown below. The remaining system prompt is the same as above.

```
def register_animation(docstring: str | None = None):
    """
    Registers an animation function which is stored in the global `animation_func`. You
    ↪   can pass an optional docstring.

    If you register a function, there a couple of rules:
        - That function should never be called anywhere else in the program. This
        ↪   function gets used later by the rendering engine.
        - This function needs a return type of `Generator[Shape, None, None]`.
    """
```

The full user prompt for 4D generation is displayed below.

```
Here are some examples of how to use helper.py:

from helper import *

"""
three ghosts chasing a yellow pacman
"""

@register()
def pacman() -> Shape:
    return primitive_call('cube', color=(1, 1, 0), scale=.8)

@register()
def ghosts() -> Shape:
    return loop(3, lambda i: transform_shape(
        library_call('ghost', color=(i / 3, 1 - i / 3, 1 - i / 3)),
        translation_matrix([i, 0, 0])
    ))

@register()
def ghost(color) -> Shape:
    return primitive_call('sphere', color=color, scale=.8)

@register_animation()
def pacman_chase_animation() -> Generator[Shape, None, None]:
    # an animated scene
    total_frames = 4  # Number of frames in the animation

    for frame in range(total_frames):
        pacman_x = - frame / total_frames
        ghost_x_offset = - 2 * frame / total_frames

        # Move pacman and ghost
        pacman = transform_shape(library_call('pacman'), translation_matrix([pacman_x, 0,
        ↪   0]))
        ghosts = transform_shape(library_call('ghosts'), translation_matrix([2 +
        ↪   ghost_x_offset, 0, 0]))

        # Export the shape, which is a frame in the animation
        yield concat_shapes(pacman, ghosts)

IMPORTANT: THE FUNCTIONS ABOVE ARE JUST EXAMPLES, YOU CANNOT USE THEM IN YOUR PROGRAM!

Now, write a similar program for the given task:

from helper import *

"""
{task}
"""
```

### E.4 MINECRAFT RENDERING

To prompt LM to generate Minecraft-compatible outputs, we remove `rotation_matrix` and `reflection_matrix` from the system prompt in Appendix E.1 and change the function header for `primitive_call` to the follows:

```python
def primitive_call(name: Literal['set_cuboid', 'delete_blocks'], **kwargs) -> Shape:
    """
    Args:
        name: str - the name of the primitive action
            support 'set_cuboid', 'delete_blocks'
        ...: Any - additional arguments for the primitive action
            For 'set_cuboid':
                - block_type: a string that denotes the block type, e.g. 'oak_log'. THESE
                ↪  MUST BE VALID LITEMATIC BLOCK TYPES.
                - block_kwargs: a dict[str, str] of additional properties to define a
                ↪  block's state fully, e.g. for 'oak_log', we need to define the axis
                ↪  with possible values 'x', 'y', or 'z'
                - scale: a list of 3 elements, denoting the scaling along the positive x,
                ↪  y, and z axises respectively.  IMPORTANT: THESE CAN ONLY BE INTEGERS!
                - fill: a boolean, describing whether the cuboid should be filled, or be
                ↪  hollow. Hint: this can be useful for creating structures that should
                ↪  be hollow, such as a building.
            For 'delete_blocks':
                - scale: a list of 3 elements, denoting the scaling along the positive x,
                ↪  y, and z axises respectively.  IMPORTANT: THESE CAN ONLY BE INTEGERS!
    Returns:
        Shape -
            For 'set_cuboid': a cuboid composed of Minecraft blocks, with the closest
            ↪  block at (0, 0, 0) and furthest (right, back-most) block at (scale[0],
            ↪  scale[1], scale[2]).
            For 'delete_blocks': an empty cuboid-shaped space without any blocks,
            ↪  starting from the closest block at (0, 0, 0) and furthest (right,
            ↪  back-most) block at (scale[0], scale[1], scale[2]).
    """
```

And we change the example program for user prompt accordingly to the follows:

```python
from helper import *

"""
A red cube on the top left of a blue pyramid of height 4.
"""

@register()
def cube_set() -> Shape:
    return concat_shapes(
        library_call('red_cube'),  # expects a cube with left-bottom-front corner block
        ↪  at (-2, 7, 2) and dims 2x2x2
        library_call('blue_pyramid'),  # expects a blue pyramid of height 4
    )  # hint: these library calls must be implemented to be compatible with the usage

@register()
def red_cube() -> Shape:
    return transform_shape(
        primitive_call('set_cuboid', block_type='minecraft:redstone_block', scale=(2, 2,
        ↪  2), fill=True),
        translation_matrix([-2, 7, 2]))

@register()
def blue_pyramid(n: int = 4) -> Shape:
    def create_pyramid_layer(i):
        # Logic here is that for the ith layer, it has dims (2*i + 1) x1x(2*i + 1.
        # We need to then shift that in the x dimension to center it, and then also in
        ↪  the y dimension to lift to the right layer of the pyramid.
        side_length = i * 2 + 1
        last_layer_length = n * 2 + 1
        x_z_offset = (last_layer_length - side_length) // 2
        y_offset = n - i - 1
        return transform_shape(
            primitive_call('set_cuboid', block_type='minecraft:lapis_block',
            ↪  scale=(side_length, 1, side_length),
                           fill=True),
            translation_matrix([x_z_offset, y_offset, x_z_offset]))

    return loop(4, create_pyramid_layer)

"""
A forest of trees of varying heights.
"""
```

```
@register()
def forest(leaf_size: int = 3) -> Shape:
    # Double for loop for placing the trees
    tree_padding = leaf_size * 2 + 3   # This is how far the center point of each tree
    ↪   should be from each other
    return loop(4, lambda i: loop(4, lambda j:
    transform_shape(library_call('simple_tree', height=random.randint(3, 7)), # Make it
    ↪   random to give the appearance of having varying heights
                    translation_matrix([i * leaf_size + tree_padding, 0, j * leaf_size +
                    ↪   tree_padding]))))

@register()
def simple_tree(height: int = 4) -> Shape:
    return concat_shapes(
        library_call('trunk', trunk_height=height),
        transform_shape(library_call('leaves', leaf_size=3), # If you pass in extra
        ↪   arguments to library_call, they need to be NAMED arguments. Passing in 3 here
        ↪   without "leaf_size" will error.
                        translation_matrix([-1, height, -1])  # Center the leaves on top
                        ↪   of the trunk
                        ))

@register()
def leaves(leaf_size: int = 3) -> Shape:
    return primitive_call('set_cuboid', block_type='minecraft:oak_leaves',
    ↪   block_kwargs={'distance': '7', 'persistent': "true", 'waterlogged': "false"},
    ↪   scale=(leaf_size, leaf_size, leaf_size), fill=True)

@register()
def trunk(trunk_height: int = 4) -> Shape:
    return primitive_call('set_cuboid', block_type='minecraft:oak_log',
    ↪   block_kwargs={'axis': 'y'}, scale=(1, trunk_height, 1), fill=True)
```

# F EXPERIMENT DETAILS

## F.1 TEXTUAL INVERSION OPTIMIZATION

To obtain image-conditioned embedding, we follow the procedure proposed in Gal et al. (2023). For the input image $I$ and text prompt $c_j$, we first use $c_j$ as guidance of GroundingSAM to obtain the desired mask of the corresponding entity. The cropped region is pad to square and resized to desired resolution, resulting in image target $I_j$. The background of $I_j$ is set to random grayscale color as used in Shi et al. (2023).

We adopt the language template "`<cls>`, 3d model, in the style of `<style>`" in all the textual inversion experiments. The template is first converted into token embeddings, then using CLIP text-encoder $g_{\text{CLIP}}$ to transform to embeddings $z_j$ for diffusion model $\hat{\epsilon}_{\text{MVDream}}$. In each textual-inversion iteration, we optimize the token embeddings $v_{j1}, v_{j2}$ for `<cls>` and `<style>` while freezing others. We use the similar objective as in diffusion model training:

$$v_{j1}^*, v_{j2}^* = \arg\min_{v_{j1}, v_{j2}} \mathbb{E}_{\eta \sim \mathcal{U}(0,1), \epsilon \sim \mathcal{N}(0,1)} \left[ \| \epsilon - \hat{\epsilon}_{\text{MVDream}}(\alpha_\eta I_j + \alpha_\eta \epsilon, \eta, z_j(v_{j1}, v_{j2})) \|_2^2 \right].$$

For each entity, we optimize the corresponding embeddings for 100 iterations with learning rate 1e-2. Empirically we find this setting is enough to fit the image conditions. After textual inversion, the embedding $z_j$ is computed with optimized token embeddings, and used to guide the entity optimization as explaint in Appendix D.

## F.2 GRAPHDREAMER IMPLEMENTATION

Since the original paper didn't release the script for automatic scene graph generation, we follow the descriptions in the paper and re-implement this stage to query LM to output scene graphs in json format to avoid manually converting LM outputs to model configurations. The full system prompt is shown below:

```
You are helpful agent and can only write output wrapped in json.
```

The full user prompt is shown below, where the given example input and output are taken from the teaser figure of the original paper (Gao et al., 2024). In below, {task} is a placeholder for input text prompts of scenes.

```
Please follow the examples in the Visual Genome dataset and generate a scene graph in
json format that best describes an input text.  The output must contain four fields:
"scene", "nodes", "edges", and "attributes".

        • "scene" is the description of the input scene.

        • "nodes" is a list of objects in the scene.  Maximum is three objects.

        • "edges" is a cyclic list of relationships between objects.  Namely, each edge is
          a list of three elements:  [object1, relationship, object2], where object1 and
          object2 are in the "nodes" list.  The number of edges must be no more than number
          of possible pairs of objects in the "nodes" list.

        • "attributes" is a dictionary where each key is an object in the "nodes" list and
          the value is a list of its attributes.

Exampl input:

A Wizard standing in front of a Wooden Desk, gazing into a Crystal Ball placed on the
Wooden Desk, with a Stack of Ancient Spell Books sitting on the Wooden Desk and next to
the crystal ball.

Example output:

{
    "scene": "A Wizard standing in front of a Wooden Desk, gazing into a Crystal Ball
    ↪  placed on the Wooden Desk, with a Stack of Ancient Spell Books sitting on the
    ↪  Wooden Desk and next to the crystal ball.",
    "nodes": ["Wizard", "Wooden Desk", "Crystal Ball", "Stack of Ancient Spell Books"],
    "edges": [
        ["Wizard", "standing in front of", "Wooden Desk"],
        ["Crystal Ball", "placed on", "Wooden Desk"],
        ["Wizard", "gazing into", "Crystal Ball"],
        ["Stack of Ancient Spell Books", "sitting on", "Wooden Desk"],
        ["Stack of Ancient Spell Books", "next to", "Crystal Ball"]
    ],
    "attributes": {
        "Wizard": ["bearded", "robed", "wise-looking"],
        "Crystal Ball": ["small", "glowing", "transparent", "mystical"],
        "Stack of Ancient Spell Books": ["leather-bound", "weathered", "rune-etched"],
        "Wooden Desk": ["sturdy", "carved", "aged"]
}

Your input:

{task}

Your output starts here.  The output must be in a valid json format.
```

After generating scene graphs with aforementioned prompt, we follow the released implementation to run Graphdreamer optimization. Each object in nodes list is represented as an individual object SDF and color field, and adopts Instant-NGP (Müller et al., 2022) as the backbone. The SDS loss optimizes not only each object, but also object pairs described in edges list. Note that Graphdreamer didn't explicitly model the repetition of objects when dealing with scenes containing repetitive components. As a result, when the raw scene graph output contains too many objects, we'll rerun the graph generation and add "The maximum number of objects is three." in text prompt to avoid reaching memory limitation during optimization.

### F.3 USER STUDY DETAILS

We provide the following instruction in the user study: "In the following questions, you'll be provided a list of text descriptions and corresponding three generation results. Please choose the one that has best text-alignment, taking into account both the visual quality and the adherence to the layout structure outlined in the text prompt." For each of the 9 scenes being evaluated, we ask the user to choose one of the three video rendering results, generated by our method and two baselines in randomized order. All videos shown in the user study are displayed on the website in Appendix B.

## G LANGUAGE MODEL OUTPUTS

### G.1 TEXT-CONDITIONED 3D GENERATION

**Example 1.** This example corresponds to Fig. 1.

Input: ``Moai in Ahu Akivi, with slight variations''. Output:

```
Here's a program to create Moai statues in Ahu Akivi with slight variations:

from helper import *

"""
Moai in Ahu Akivi, with slight variations
"""

@register()
def moai_head(scale: float) -> Shape:
    head = primitive_call('cube', color=(0.5, 0.5, 0.5), shape_kwargs={'scale': (scale,
    ↪   scale*1.5, scale*0.8)})
    nose = primitive_call('cube', color=(0.45, 0.45, 0.45), shape_kwargs={'scale':
    ↪   (scale*0.2, scale*0.4, scale*0.3)})
    nose = transform_shape(nose, translation_matrix((0, -scale*0.2, scale*0.4)))
    return concat_shapes(head, nose)

@register()
def moai_body(scale: float) -> Shape:
    body = primitive_call('cube', color=(0.55, 0.55, 0.55), shape_kwargs={'scale':
    ↪   (scale*0.8, scale*2, scale*0.6)})
    return body

@register()
def moai(scale: float) -> Shape:
    head = library_call('moai_head', scale=scale)
    body = library_call('moai_body', scale=scale)

    head_min = compute_shape_min(head)
    body_max = compute_shape_max(body)

    head = transform_shape(head, translation_matrix((0, body_max[1] - head_min[1], 0)))

    return concat_shapes(body, head)

@register()
def ahu_platform(width: float, height: float, depth: float) -> Shape:
    platform = primitive_call('cube', color=(0.3, 0.3, 0.3), shape_kwargs={'scale':
    ↪   (width, height, depth)})
    return platform

@register()
def moai_with_variation(base_scale: float) -> Shape:
    scale_variation = np.random.uniform(0.9, 1.1)
    rotation_variation = np.random.uniform(-0.05, 0.05)

    moai_statue = library_call('moai', scale=base_scale * scale_variation)
    moai_center = compute_shape_center(moai_statue)

    return transform_shape(moai_statue, rotation_matrix(rotation_variation, (0, 1, 0),
    ↪   moai_center))

@register()
def ahu_akivi_scene() -> Shape:
    platform = library_call('ahu_platform', width=14, height=1, depth=4)
    platform_top = compute_shape_max(platform)[1]

    def create_moai(i: int) -> Shape:
        moai = library_call('moai_with_variation', base_scale=1.5)
        moai_bottom = compute_shape_min(moai)[1]
        x_position = (i - 3) * 2  # Distribute 7 Moai evenly
        return transform_shape(moai, translation_matrix((x_position, platform_top -
        ↪   moai_bottom, 0)))

    moai_statues = loop(7, create_moai)

    return concat_shapes(platform, moai_statues)
```

This program creates a scene of Moai statues in Ahu Akivi with the following components:

  1. moai_head:  Creates a simplified Moai head using cubes for the main head and
     nose.
  2. moai_body:  Creates a simplified Moai body using a cube.
  3. moai:  Combines the head and body to form a complete Moai statue.
  4. ahu_platform:  Creates the stone platform (ahu) on which the Moai stand.
  5. moai_with_variation:  Applies slight variations in scale and rotation to each
     Moai.
  6. ahu_akivi_scene:  Creates the entire scene with a platform and seven Moai statues.

The scene features seven Moai statues standing on a stone platform, similar to the real
Ahu Akivi site.  Each Moai has slight variations in size and rotation to add visual
interest and realism to the scene.

**Example 2.** This example corresponds to Fig. 2.

Input: ``a chessboard at game start''. Output:

```
Here's a program to create a chessboard at game start using the provided helper.py:

from helper import *

"""
a chessboard at game start
"""

@register("chess_square")
def chess_square(color: tuple[float, float, float]) -> Shape:
    return primitive_call('cube', color=color, shape_kwargs={'scale': (0.125, 0.02,
    ↪  0.125)})

@register("chessboard")
def chessboard() -> Shape:
    def loop_fn(i) -> Shape:
        row = i // 8
        col = i % 8
        color = (0.9, 0.9, 0.9) if (row + col) % 2 == 0 else (0.1, 0.1, 0.1)
        square = library_call('chess_square', color=color)
        return transform_shape(square, translation_matrix((col * 0.125, 0, row * 0.125)))

    return loop(64, loop_fn)

@register("chess_pawn")
def chess_pawn(color: tuple[float, float, float]) -> Shape:
    base = primitive_call('cylinder', color=color, shape_kwargs={'radius': 0.05, 'p0':
    ↪  (0, 0, 0), 'p1': (0, 0.02, 0)})
    body = primitive_call('cylinder', color=color, shape_kwargs={'radius': 0.04, 'p0':
    ↪  (0, 0.02, 0), 'p1': (0, 0.08, 0)})
    head = primitive_call('sphere', color=color, shape_kwargs={'radius': 0.035})
    head = transform_shape(head, translation_matrix((0, 0.115, 0)))
    return concat_shapes(base, body, head)

@register("chess_rook")
def chess_rook(color: tuple[float, float, float]) -> Shape:
    base = primitive_call('cylinder', color=color, shape_kwargs={'radius': 0.05, 'p0':
    ↪  (0, 0, 0), 'p1': (0, 0.02, 0)})
    body = primitive_call('cylinder', color=color, shape_kwargs={'radius': 0.04, 'p0':
    ↪  (0, 0.02, 0), 'p1': (0, 0.1, 0)})
    top = primitive_call('cube', color=color, shape_kwargs={'scale': (0.09, 0.03, 0.09)})
    top = transform_shape(top, translation_matrix((0, 0.115, 0)))
    return concat_shapes(base, body, top)

@register("chess_knight")
def chess_knight(color: tuple[float, float, float]) -> Shape:
    base = primitive_call('cylinder', color=color, shape_kwargs={'radius': 0.05, 'p0':
    ↪  (0, 0, 0), 'p1': (0, 0.02, 0)})
    body = primitive_call('cylinder', color=color, shape_kwargs={'radius': 0.04, 'p0':
    ↪  (0, 0.02, 0), 'p1': (0, 0.08, 0)})
    head = primitive_call('cube', color=color, shape_kwargs={'scale': (0.06, 0.08,
    ↪  0.04)})
    head = transform_shape(head, translation_matrix((0.01, 0.11, 0)))
    head = transform_shape(head, rotation_matrix(0.3, (0, 0, 1), (0, 0.11, 0)))
    return concat_shapes(base, body, head)

@register("chess_bishop")
def chess_bishop(color: tuple[float, float, float]) -> Shape:
    base = primitive_call('cylinder', color=color, shape_kwargs={'radius': 0.05, 'p0':
    ↪  (0, 0, 0), 'p1': (0, 0.02, 0)})
    body = primitive_call('cylinder', color=color, shape_kwargs={'radius': 0.04, 'p0':
    ↪  (0, 0.02, 0), 'p1': (0, 0.1, 0)})
    head = primitive_call('sphere', color=color, shape_kwargs={'radius': 0.03})
    head = transform_shape(head, translation_matrix((0, 0.13, 0)))
    top = primitive_call('cylinder', color=color, shape_kwargs={'radius': 0.01, 'p0': (0,
    ↪  0.13, 0), 'p1': (0, 0.16, 0)})
    return concat_shapes(base, body, head, top)

@register("chess_queen")
def chess_queen(color: tuple[float, float, float]) -> Shape:
    base = primitive_call('cylinder', color=color, shape_kwargs={'radius': 0.055, 'p0':
    ↪  (0, 0, 0), 'p1': (0, 0.02, 0)})
    body = primitive_call('cylinder', color=color, shape_kwargs={'radius': 0.045, 'p0':
    ↪  (0, 0.02, 0), 'p1': (0, 0.12, 0)})
    crown = primitive_call('sphere', color=color, shape_kwargs={'radius': 0.04})
    crown = transform_shape(crown, translation_matrix((0, 0.16, 0)))
    top = primitive_call('cylinder', color=color, shape_kwargs={'radius': 0.01, 'p0': (0,
    ↪  0.16, 0), 'p1': (0, 0.2, 0)})
    return concat_shapes(base, body, crown, top)

@register("chess_king")
def chess_king(color: tuple[float, float, float]) -> Shape:
    base = primitive_call('cylinder', color=color, shape_kwargs={'radius': 0.06, 'p0':
    ↪  (0, 0, 0), 'p1': (0, 0.02, 0)})
```

```
      body = primitive_call('cylinder', color=color, shape_kwargs={'radius': 0.05, 'p0':
      ↪  (0, 0.02, 0), 'p1': (0, 0.14, 0)})
      crown = primitive_call('cube', color=color, shape_kwargs={'scale': (0.1, 0.03, 0.1)})
      crown = transform_shape(crown, translation_matrix((0, 0.17, 0)))
      cross_v = primitive_call('cube', color=color, shape_kwargs={'scale': (0.02, 0.06,
      ↪  0.02)})
      cross_v = transform_shape(cross_v, translation_matrix((0, 0.215, 0)))
      cross_h = primitive_call('cube', color=color, shape_kwargs={'scale': (0.06, 0.02,
      ↪  0.02)})
      cross_h = transform_shape(cross_h, translation_matrix((0, 0.215, 0)))
      return concat_shapes(base, body, crown, cross_v, cross_h)

@register("chess_pieces")
def chess_pieces() -> Shape:
    white = (0.9, 0.9, 0.9)
    black = (0.1, 0.1, 0.1)
    pieces = []

    # Place pawns
    for i in range(8):
        white_pawn = library_call('chess_pawn', color=white)
        black_pawn = library_call('chess_pawn', color=black)
        pieces.append(transform_shape(white_pawn, translation_matrix((i * 0.125, 0.02,
        ↪  0.125))))
        pieces.append(transform_shape(black_pawn, translation_matrix((i * 0.125, 0.02,
        ↪  0.75))))

    # Place other pieces
    piece_order = ['rook', 'knight', 'bishop', 'queen', 'king', 'bishop', 'knight',
    ↪  'rook']
    for i, piece in enumerate(piece_order):
        white_piece = library_call(f'chess_{piece}', color=white)
        black_piece = library_call(f'chess_{piece}', color=black)
        pieces.append(transform_shape(white_piece, translation_matrix((i * 0.125, 0.02,
        ↪  0))))
        pieces.append(transform_shape(black_piece, translation_matrix((i * 0.125, 0.02,
        ↪  0.875))))

    return concat_shapes(*pieces)

@register("chessboard_at_game_start")
def chessboard_at_game_start() -> Shape:
    board = library_call('chessboard')
    pieces = library_call('chess_pieces')
    return concat_shapes(board, pieces)

This program creates a chessboard at game start with the following components:

    1. A chessboard made of alternating white and black squares.
    2. Chess pieces for both white and black sides, including pawns, rooks, knights,
       bishops, queens, and kings.
    3. The pieces are placed in their correct starting positions on the board.

To use this program and generate the chessboard at game start, you would call:

chessboard_scene = library_call('chessboard_at_game_start')

This will return a complete chessboard with all pieces in their starting positions.
```

## G.2 TEXT-CONDITIONED 3D EDITING

In this task, for each example, we prompt LM in two rounds—the first round with a scene description as task specification, and the second round with editing instructions. LM outputs in the first round are omitted.

**Example 1.** Input scene description: ``Rene Magritte The Son of Man''. Input editing instruction: ``Move the apple to the left''. Output:

```
Here's the minimally edited program that moves the apple to the left:

from helper import *

"""
Move the apple to left
"""

@register()
def suit(width: float, height: float, depth: float) -> Shape:
    return primitive_call('cube', color=(0.1, 0.1, 0.1), shape_kwargs={'scale': (width,
    ↪  height, depth)})
```

```
1404
1405        @register()
1406        def head(radius: float) -> Shape:
               return primitive_call('sphere', color=(0.9, 0.75, 0.65), shape_kwargs={'radius':
1407           ↪   radius})

1408        @register()
1409        def apple(radius: float) -> Shape:
               return primitive_call('sphere', color=(0.8, 0.1, 0.1), shape_kwargs={'radius':
1410           ↪   radius})

1411        @register()
1412        def hat(radius: float, height: float) -> Shape:
               brim = primitive_call('cylinder', color=(0.1, 0.1, 0.1), shape_kwargs={'radius':
1413           ↪   radius * 1.2, 'p0': (0, 0, 0), 'p1': (0, height * 0.1, 0)})
               top = primitive_call('cylinder', color=(0.1, 0.1, 0.1), shape_kwargs={'radius':
1414           ↪   radius, 'p0': (0, height * 0.1, 0), 'p1': (0, height, 0)})
               return concat_shapes(brim, top)
1415
1416        @register()
           def son_of_man() -> Shape:
1417           # Create the suit
               suit_shape = library_call('suit', width=0.6, height=1.0, depth=0.3)
1418
1419           # Create the head
               head_shape = library_call('head', radius=0.15)
1420           head_shape = transform_shape(head_shape, translation_matrix((0, 0.6, 0)))

1421           # Create the apple
1422           apple_shape = library_call('apple', radius=0.08)
               apple_shape = transform_shape(apple_shape, translation_matrix((-0.15, 0.62, 0.15)))
1423           ↪   # Changed x-coordinate to -0.15

1424           # Create the hat
1425           hat_shape = library_call('hat', radius=0.12, height=0.15)
               hat_shape = transform_shape(hat_shape, translation_matrix((0, 0.78, 0)))
1426
1427           # Combine all shapes
               figure = concat_shapes(suit_shape, head_shape, apple_shape, hat_shape)
1428
1429           # Add a slight tilt to the figure
               figure_center = compute_shape_center(figure)
1430           tilted_figure = transform_shape(figure, rotation_matrix(0.1, (1, 0, 0),
               ↪   figure_center))
1431
1432           return tilted_figure

1433    The only change made to the program is in the son_of_man() function.  Specifically, the
1434    translation of the apple shape has been modified.  The x-coordinate in the translation
        matrix for the apple has been changed from 0 to -0.15:
1435
1436    apple_shape = transform_shape(apple_shape, translation_matrix((-0.15, 0.62, 0.15)))  #
        ↪   Changed x-coordinate to -0.15
1437
1438    This change moves the apple to the left by 0.15 units in the scene's coordinate system,
1439    where the negative x-direction corresponds to the left side of the figure.  The y and z
1440    coordinates remain the same, keeping the apple at the same height and depth relative to
1441    the face.
1442
```

**Example 2.** Input scene description: ``Paul Klee Castle and Sun''. Input editing instruction: ``Change all castles to be the middle one''. Output:

```
1445    Here's the minimally edited program to change all castles to be the middle one:
1446
1447    from helper import *
1448    """
        Change all castles to be the middle one
1449    """

1450        @register()
1451        def rectangle(width: float, height: float, color: tuple[float, float, float]) -> Shape:
               return primitive_call('cube', shape_kwargs={'scale': (width, height, 0.01)},
1452           ↪   color=color)

1453        @register()
1454        def triangle(base: float, height: float, color: tuple[float, float, float]) -> Shape:
               triangle_shape = primitive_call('cube', shape_kwargs={'scale': (base, height, 0.01)},
1455           ↪   color=color)
               return transform_shape(triangle_shape, translation_matrix((0, height/2, 0)))
1456
1457        @register()
```

```
1458    def sun(radius: float) -> Shape:
1459        return primitive_call('sphere', shape_kwargs={'radius': radius}, color=(1.0, 0.8, 0))

1460    @register()
1461    def castle_block(width: float, height: float, color: tuple[float, float, float]) ->
        ↪  Shape:
1462        return library_call('rectangle', width=width, height=height, color=color)

1463    @register()
1464    def castle_tower(width: float, height: float, color: tuple[float, float, float]) ->
        ↪  Shape:
1465        tower = library_call('castle_block', width=width, height=height, color=color)
1466        roof = library_call('triangle', base=width, height=width/2, color=color)
            tower_height = compute_shape_sizes(tower)[1]
1467        return concat_shapes(
                tower,
1468            transform_shape(roof, translation_matrix((0, tower_height, 0)))
            )
1469
1470    @register()
        def castle() -> Shape:
1471        base = library_call('castle_block', width=0.8, height=0.4, color=(0.5, 0.5, 0.7))
            center_tower = library_call('castle_tower', width=0.25, height=0.7, color=(0.5, 0.5,
1472        ↪  0.7))

1473
            base_width = compute_shape_sizes(base)[0]
1474        center_tower_width = compute_shape_sizes(center_tower)[0]

1475
            return concat_shapes(
1476            base,
                transform_shape(center_tower, translation_matrix((-base_width/4, 0.4, 0))),
1477            transform_shape(center_tower, translation_matrix((base_width/4, 0.4, 0))),
                transform_shape(center_tower, translation_matrix((0, 0.4, 0)))
1478        )

1479
1480    @register()
        def background() -> Shape:
1481        return library_call('rectangle', width=2, height=2, color=(0.95, 0.95, 0.8))

1482    @register()
        def castle_and_sun() -> Shape:
1483        bg = library_call('background')
            castle_shape = library_call('castle')
1484        sun_shape = library_call('sun', radius=0.15)

1485
            return concat_shapes(
1486            bg,
                transform_shape(castle_shape, translation_matrix((0, -0.3, 0.01))),
1487            transform_shape(sun_shape, translation_matrix((0.5, 0.5, 0.02)))
            )
1488
1489
        In this edited version, I made the following changes to fulfill the instruction "Change
1490    all castles to be the middle one":

1491
            1. In the castle() function, I removed the left_tower and right_tower variables.
1492        2. I changed the base color to match the center tower's color: (0.5, 0.5, 0.7).
            3. I replaced the left and right towers with two additional center towers.
1493        4. I adjusted the positioning of the three center towers to be evenly spaced across
                the base.
1494
1495
1496    Now, when the castle_and_sun() function is called, it will create a scene where all
1497    the castle towers are identical to the original middle tower, fulfilling the given
1498    instruction.
1499
```

**Example 3.** Input scene description: ``Bramante Staircase, Vatican Museums''.
Input editing instruction: ``shrink staircase radius by 80%''. Output:

```
1502    Here's the minimally edited program that shrinks the staircase radius by 80%:
1503
        from helper import *
1504
        """
1505    Shrink staircase radius by 80%
        """
1506
1507    @register()
        def stair_step(width: float, height: float, depth: float) -> Shape:
1508        return primitive_call('cube', color=(0.8, 0.8, 0.8), shape_kwargs={'scale': (width,
1509        ↪  height, depth)})

1510    @register()
1511    def spiral_staircase(radius: float, height: float, num_steps: int) -> Shape:
```

```
1512        def loop_fn(i: int) -> Shape:
1513            angle = (i / num_steps) * 2 * math.pi
1514            step_width = 0.8 * radius
             step_height = height / num_steps
1515            step_depth = 0.3 * radius

1516            step = library_call('stair_step', width=step_width, height=step_height,
1517            ↪  depth=step_depth)

1518            y_offset = i * step_height
             x_offset = radius * math.cos(angle)
1519            z_offset = radius * math.sin(angle)

1520            step = transform_shape(step, translation_matrix((x_offset, y_offset, z_offset)))
1521            return transform_shape(step, rotation_matrix(angle, (0, 1, 0), (x_offset,
             ↪  y_offset, z_offset)))
1522
1523        return loop(num_steps, loop_fn)

1524    @register()
     def central_column(radius: float, height: float) -> Shape:
1525        return primitive_call('cylinder', color=(0.7, 0.7, 0.7),
1526                              shape_kwargs={'radius': radius, 'p0': (0, 0, 0), 'p1': (0,
                              ↪  height, 0)})
1527
1528    @register()
     def bramante_staircase() -> Shape:
1529        staircase_radius = 5.0 * 0.2  # Shrink staircase radius by 80%
         staircase_height = 15.0
1530        num_steps = 60
         column_radius = 0.5 * 0.2  # Shrink column radius proportionally
1531
1532        stairs = library_call('spiral_staircase', radius=staircase_radius,
         ↪  height=staircase_height, num_steps=num_steps)
1533        column = library_call('central_column', radius=column_radius,
         ↪  height=staircase_height)
1534
1535        return concat_shapes(stairs, column)
```

```
1536    In this edited version, I've made the following changes to shrink the staircase radius by
1537    80%:
1538
1539        1. In the bramante_staircase function, I've modified the staircase_radius by
1540           multiplying it by 0.2 (which is equivalent to reducing it by 80%):
1541
                staircase_radius = 5.0 * 0.2  # Shrink staircase radius by 80%
1542
        2. To maintain proportions, I've also adjusted the column_radius by the same factor:
1543
                column_radius = 0.5 * 0.2  # Shrink column radius proportionally
1544
1545    These changes will result in a Bramante staircase that has a radius 80% smaller than the
        original, while maintaining the same height and number of steps.  The central column has
1546    also been proportionally reduced to maintain the overall look of the structure.
```

**Example 4.** Input scene description: ``a 8-layer 3-color Jenga set at game start''. Input editing instruction: ``remove 2 blocks from second top layer and the tower should not fall''. Output:

```
1551    Here's the minimally edited program to remove 2 blocks from the second top layer of the
1552    Jenga tower:

1553    from helper import *

1554    """
1555    Remove 2 blocks from second top layer and the tower should not fall
        """
1556
1557    @register()
     def jenga_block(color: tuple[float, float, float]) -> Shape:
1558        """Create a single Jenga block"""
1559        return primitive_call('cube', color=color, shape_kwargs={'scale': (0.75, 0.5, 2.25)})

1560    @register()
     def jenga_layer(is_horizontal: bool, y_offset: float, colors: list[tuple[float, float,
1561    ↪  float]], remove_blocks: list[int] = []) -> Shape:
1562        """Create a layer of 3 Jenga blocks, optionally removing specified blocks"""
         def block_fn(i: int) -> Shape:
1563            if i in remove_blocks:
                return []
1564            block = library_call('jenga_block', color=colors[i])
             if is_horizontal:
1565
```

```
                    transform = translation_matrix((i * 0.75 - 0.75, y_offset, 0))
            else:
                    transform = translation_matrix((0, y_offset, i * 0.75 - 0.75))
                    transform = transform @ rotation_matrix(math.pi/2, (0, 1, 0), (0, 0, 0))
            return transform_shape(block, transform)

        return loop(3, block_fn)

    @register()
    def jenga_tower() -> Shape:
        """Create an 8-layer Jenga tower with 2 blocks removed from the second top layer"""
        colors = [(0.8, 0.4, 0.3), (0.9, 0.7, 0.5), (0.6, 0.3, 0.1)]

        def layer_fn(i: int) -> Shape:
            is_horizontal = i % 2 == 0
            y_offset = i * 0.5
            layer_colors = [colors[i % 3], colors[(i+1) % 3], colors[(i+2) % 3]]
            remove_blocks = [0, 2] if i == 6 else []  # Remove blocks 0 and 2 from the second
            ↪  top layer (index 6)
            return library_call('jenga_layer', is_horizontal=is_horizontal,
            ↪  y_offset=y_offset, colors=layer_colors, remove_blocks=remove_blocks)

        return loop(8, layer_fn)

    @register()
    def jenga_set() -> Shape:
        """Create a complete Jenga set with 2 blocks removed from the second top layer"""
        tower = library_call('jenga_tower')
        base = primitive_call('cube', color=(0.4, 0.2, 0.1), shape_kwargs={'scale': (3, 0.1,
        ↪  3)})

        tower_min = compute_shape_min(tower)
        base_max = compute_shape_max(base)

        return concat_shapes(
            base,
            transform_shape(tower, translation_matrix((0, base_max[1] - tower_min[1], 0)))
        )
```

```
This modified version of the program removes 2 blocks from the second top layer of the
Jenga tower.  Here are the changes made:

    1. The jenga_layer function now accepts an optional remove_blocks parameter, which is
       a list of block indices to remove from the layer.

    2. In the jenga_tower function, we modify the layer_fn to remove blocks 0 and 2 (the
       leftmost and rightmost blocks) from the second top layer (index 6, since we're
       counting from 0).

    3. The rest of the program remains the same, ensuring that the tower is still placed
       on the base and the overall structure is maintained.

These changes will result in a Jenga tower with two blocks missing from the second top
layer, while the rest of the tower remains intact.  The tower should not fall because the
top layer is still fully supported by the remaining block in the second top layer.
```

## G.3 TEXT-CONDITIONED 4D GENERATION

**Example 1.** Input: ``carousel with a small canopy''. Output:

```
Here's a program that creates a carousel with a small canopy using the provided
helper.py:

from helper import *

"""
carousel with a small canopy
"""

@register()
def carousel_base() -> Shape:
    base = primitive_call('cylinder', shape_kwargs={'radius': 2, 'p0': (0, 0, 0), 'p1':
    ↪  (0, 0.2, 0)}, color=(0.6, 0.4, 0.2))
    return base

@register()
def carousel_platform() -> Shape:
    platform = primitive_call('cylinder', shape_kwargs={'radius': 1.8, 'p0': (0, 0.2, 0),
    ↪  'p1': (0, 0.3, 0)}, color=(0.8, 0.8, 0.8))
    return platform

@register()
def carousel_pole() -> Shape:
```

```python
    pole = primitive_call('cylinder', shape_kwargs={'radius': 0.1, 'p0': (0, 0.3, 0),
    ↪   'p1': (0, 3, 0)}, color=(0.7, 0.7, 0.7))
    return pole

@register()
def carousel_horse() -> Shape:
    body = primitive_call('cylinder', shape_kwargs={'radius': 0.2, 'p0': (0, 0, -0.4),
    ↪   'p1': (0, 0, 0.4)}, color=(1, 1, 1))
    head = primitive_call('sphere', shape_kwargs={'radius': 0.15}, color=(1, 1, 1))
    head = transform_shape(head, translation_matrix((0, 0.2, -0.5)))
    legs = loop(4, lambda i: transform_shape(
        primitive_call('cylinder', shape_kwargs={'radius': 0.05, 'p0': (0, 0, 0), 'p1':
        ↪   (0, -0.5, 0)}, color=(1, 1, 1)),
        translation_matrix((0.15 * (-1 if i % 2 == 0 else 1), -0.2, 0.3 * (-1 if i < 2
        ↪   else 1)))
    ))
    horse = concat_shapes(body, head, legs)
    return transform_shape(horse, translation_matrix((0, 1, 0)))

@register()
def carousel_horses() -> Shape:
    return loop(4, lambda i: transform_shape(
        library_call('carousel_horse'),
        rotation_matrix(i * math.pi / 2, (0, 1, 0), (0, 0, 0)) @ translation_matrix((1.5,
        ↪   0, 0))
    ))

@register()
def canopy() -> Shape:
    cone = primitive_call('cylinder', shape_kwargs={'radius': 2, 'p0': (0, 3, 0), 'p1':
    ↪   (0, 3.5, 0)}, color=(1, 0.5, 0.5))
    top = primitive_call('sphere', shape_kwargs={'radius': 0.1}, color=(1, 0.7, 0.7))
    top = transform_shape(top, translation_matrix((0, 3.6, 0)))
    return concat_shapes(cone, top)

@register()
def carousel() -> Shape:
    base = library_call('carousel_base')
    platform = library_call('carousel_platform')
    pole = library_call('carousel_pole')
    horses = library_call('carousel_horses')
    canopy_shape = library_call('canopy')
    return concat_shapes(base, platform, pole, horses, canopy_shape)

@register_animation()
def carousel_animation() -> Generator[Shape, None, None]:
    total_frames = 60
    for frame in range(total_frames):
        angle = frame * 2 * math.pi / total_frames
        rotated_carousel = transform_shape(
            library_call('carousel'),
            rotation_matrix(angle, (0, 1, 0), (0, 0, 0))
        )
        yield rotated_carousel
```

This program creates a carousel with a small canopy. Here's a breakdown of the components:

```
    1. carousel_base:  A cylindrical base for the carousel.
    2. carousel_platform:  A slightly smaller cylinder on top of the base.
    3. carousel_pole:  A central pole for the carousel.
    4. carousel_horse:  A simple horse shape made from cylinders and a sphere.
    5. carousel_horses:  Four horses arranged around the central pole.
    6. canopy:  A small conical canopy with a spherical top.
    7. carousel:  Combines all the components to create the complete carousel.
    8. carousel_animation:  An animation that rotates the carousel.
```

The carousel is designed with a modular approach, allowing easy modifications and adjustments to individual components. The animation function rotates the entire carousel, simulating its movement.

**Example 2.** Input: ``solar system model''. Output:

```
Here's a program that creates a simple solar system model using the provided helper.py:

from helper import *

"""
solar system model
"""

@register()
```

```
1674    def sun() -> Shape:
1675        return primitive_call('sphere', shape_kwargs={'radius': 0.5}, color=(1, 0.7, 0))
1676
        @register()
1677    def planet(radius: float, color: tuple[float, float, float]) -> Shape:
            return primitive_call('sphere', shape_kwargs={'radius': radius}, color=color)
1678
        @register()
1679    def orbit(radius: float) -> Shape:
1680        points = 100
            orbit_shape = []
1681        for i in range(points):
1682            angle = 2 * math.pi * i / points
                x = radius * math.cos(angle)
1683            z = radius * math.sin(angle)
                orbit_point = primitive_call('sphere', shape_kwargs={'radius': 0.01}, color=(0.5,
1684            ↪  0.5, 0.5))
                orbit_shape.append(transform_shape(orbit_point, translation_matrix((x, 0, z))))
1685        return concat_shapes(*orbit_shape)
1686
        @register()
1687    def solar_system() -> Shape:
1688        sun_shape = library_call('sun')
            mercury_orbit = library_call('orbit', radius=1.5)
1689        venus_orbit = library_call('orbit', radius=2.5)
            earth_orbit = library_call('orbit', radius=3.5)
1690        mars_orbit = library_call('orbit', radius=4.5)
1691        return concat_shapes(
1692            sun_shape,
                mercury_orbit,
1693            venus_orbit,
                earth_orbit,
1694            mars_orbit
1695        )
1696    @register_animation()
        def solar_system_animation() -> Generator[Shape, None, None]:
1697        total_frames = 120
1698        for frame in range(total_frames):
1699            t = frame / total_frames
1700            sun = library_call('sun')
1701            mercury = transform_shape(
1702                library_call('planet', radius=0.1, color=(0.7, 0.7, 0.7)),
                    rotation_matrix(t * 4 * math.pi, (0, 1, 0), (0, 0, 0)) @
1703                ↪  translation_matrix((1.5, 0, 0))
1704            )
1705            venus = transform_shape(
                    library_call('planet', radius=0.15, color=(0.9, 0.7, 0.4)),
1706                rotation_matrix(t * 3 * math.pi, (0, 1, 0), (0, 0, 0)) @
1707                ↪  translation_matrix((2.5, 0, 0))
            )
1708
                earth = transform_shape(
1709                library_call('planet', radius=0.17, color=(0.2, 0.4, 0.8)),
1710                rotation_matrix(t * 2 * math.pi, (0, 1, 0), (0, 0, 0)) @
                    ↪  translation_matrix((3.5, 0, 0))
1711            )
1712            mars = transform_shape(
1713                library_call('planet', radius=0.13, color=(0.8, 0.3, 0.1)),
                    rotation_matrix(t * 1.5 * math.pi, (0, 1, 0), (0, 0, 0)) @
1714                ↪  translation_matrix((4.5, 0, 0))
1715            )
1716            solar_system = library_call('solar_system')
1717            yield concat_shapes(solar_system, sun, mercury, venus, earth, mars)
1718
    This program creates a simple solar system model with the following components:
1719
1720        1. A sun at the center.
1721        2. Four planets (Mercury, Venus, Earth, and Mars) orbiting around the sun.
            3. Orbit paths for each planet.
1722
1723    The solar_system_animation function generates an animation of the planets orbiting around
1724    the sun. Each planet moves at a different speed, simulating their relative orbital
        periods.
1725
1726    Here's a breakdown of the functions:
1727
```

- `sun()`: Creates a yellow sphere representing the sun.
- `planet(radius, color)`: Creates a sphere with the given radius and color, representing a planet.
- `orbit(radius)`: Creates a circular orbit path at the given radius.
- `solar_system()`: Combines the sun and orbit paths into a static solar system model.
- `solar_system_animation()`: Animates the planets orbiting around the sun.

The animation runs for 120 frames, with each planet completing a different fraction of its orbit in that time. The sun remains stationary at the center while the planets rotate around it.

**Example 3.** Input: ``a toy wind turbine''. Output:

Here's a program to create a toy wind turbine using the provided helper.py:

```python
from helper import *

"""
a toy wind turbine
"""

@register()
def turbine_base() -> Shape:
    base = primitive_call('cylinder', shape_kwargs={'radius': 0.2, 'p0': (0, 0, 0), 'p1':
    ↪  (0, 2, 0)}, color=(0.5, 0.5, 0.5))
    return base

@register()
def turbine_nacelle() -> Shape:
    nacelle = primitive_call('cube', shape_kwargs={'scale': (0.6, 0.4, 0.8)}, color=(0.7,
    ↪  0.7, 0.7))
    return transform_shape(nacelle, translation_matrix((0, 2.2, 0)))

@register()
def turbine_blade() -> Shape:
    blade = primitive_call('cube', shape_kwargs={'scale': (0.1, 1.5, 0.05)}, color=(1, 1,
    ↪  1))
    return transform_shape(blade, translation_matrix((0, 0.75, 0)))

@register()
def turbine_rotor() -> Shape:
    return loop(3, lambda i: transform_shape(
        library_call('turbine_blade'),
        rotation_matrix(i * 2 * math.pi / 3, (0, 0, 1), (0, 0, 0))
    ))

@register()
def wind_turbine() -> Shape:
    base = library_call('turbine_base')
    nacelle = library_call('turbine_nacelle')
    rotor = transform_shape(library_call('turbine_rotor'), translation_matrix((0, 2.2,
    ↪  0.4)))
    return concat_shapes(base, nacelle, rotor)

@register_animation()
def wind_turbine_animation() -> Generator[Shape, None, None]:
    total_frames = 60

    for frame in range(total_frames):
        angle = frame * 2 * math.pi / total_frames

        base = library_call('turbine_base')
        nacelle = library_call('turbine_nacelle')
        rotor = transform_shape(
            library_call('turbine_rotor'),
            rotation_matrix(angle, (0, 0, 1), (0, 2.2, 0.4)) @ translation_matrix((0,
            ↪  2.2, 0.4))
        )

        yield concat_shapes(base, nacelle, rotor)
```

This program creates a toy wind turbine with the following components:

1. turbine_base: A cylindrical base for the turbine.
2. turbine_nacelle: A cuboid representing the nacelle (housing) at the top of the turbine.
3. turbine_blade: A single blade of the turbine.
4. turbine_rotor: Three blades arranged in a rotor configuration.
5. wind_turbine: The complete wind turbine, combining all the components.

The `wind_turbine_animation` function creates an animation of the turbine with rotating blades. It generates 60 frames, rotating the rotor by a full 360 degrees over the course of the animation.

