# OpenReview forum: "The Scene Language: Representing Scenes with Programs, Words, and Embeddings"
_ICLR.cc/2025/Conference — ICLR 2025 Conference Withdrawn Submission_

### Official Review · Reviewer_K12Q · 2024-11-01

**Soundness:** 2
**Presentation:** 3
**Contribution:** 2
**Rating:** 3
**Confidence:** 3

**Summary:**

The paper proposes a visual scene representation paradigm named Scene Language, which involves an ensemble of programs, words, and embeddings to describe the structure, semantics, and identity of visual scenes. A training-free inference technique is developed to infer the proposed scene representation from pre-trained language models. And a generic rendering module is utilized to render the scene into images using traditional, neural, or hybrid graphics renderers. Experimental results show that the proposed Scene Language generates complex scenes with higher fidelity while explicitly modeling the scene structures to enable precise control and editing.

**Strengths:**

1 The proposed method is applicable to multiple downstream scene generation/editing tasks. It is a training-free approach with the capability to generate and edit 3D and 4D scenes.

2 The visualization results show the superiority of the proposed method over competitors.

3 The paper is well-written with figures and tables nicely presented.

**Weaknesses:**

1 My biggest concern is that the manuscript seems to have incremental novelty since the proposed method relies on most existing models/tools and uses them straightforwardly. For example, the proposed scene representation is constructed directly using pretrained language models and is used for scene rendering using the existing rendering methods, e.g., 3D Gaussians.

2 Moreover, the proposed Scene Language looks more like an engineering improvement on GraphDreamer, breaking down complex scenes into independent entities for generation. The newly introduced embedding term also appears to have little effect. For example, in Figure 1, the referenced object should be made of metal material, while the modified object only captures information of blue color.

3 The pretrained language models are usually not ready for addressing specific downstream tasks and may produce inaccurate answers. However, the proposed method does not seem to incorporate some adaptation modules and give deep analysis.

4 The experiments also incorporate insufficient comparisons: (1) The authors should also report the FLOPs over the GraphDreamer for comprehensive comparisons. (2) If the provided image cannot describe the entire scene (e.g., occlusion), Scene Language cannot program the entire scene. Thus the comparisons to image-to-3D approaches should be included. (3) This paper lacks the ablation study on the proposal terms of descriptions on scene.

**Questions:**

See details in the weaknesses.

---

### Official Review · Reviewer_fKvz · 2024-11-03

**Soundness:** 2
**Presentation:** 3
**Contribution:** 2
**Rating:** 5
**Confidence:** 3

**Summary:**

The research paper introduces a new scene language representation designed to provide detailed information about a scene. This representation operates on three levels: programs that define the scene's composition and structural relationships between objects in the scene, semantic words that provide objects present in the scenes, and feature embeddings that capture instance-specific properties. These different layers provide a framework to describe a scene completely. They use this representation for scene generation and editing during rendering.

**Strengths:**

The paper is well-written, with a clear motivation and a thorough description of the proposed method. The application of scene editing is an important problem, highlighting the practical importance of the approach.

**Weaknesses:**

1) The examples presented in the paper seem relatively simple. Including more realistic, real-world scenes could better demonstrate the effectiveness of the scene language representation in handling complexity. For instance, incorporating indoor or outdoor scenes with multiple objects and occlusions would provide a valuable setup to showcase the approach’s robustness and highlight its results in more challenging conditions.
2) Quantifying the correctness or accuracy of the scene language generation step would add valuable insight. Specifically, it would be beneficial to measure various aspects of the scene language generation process, as outlined in Section 5.

3) It would also be valuable to demonstrate how noise and errors in the scene language are managed during generation. For instance, if the scene language contains inaccuracies, such as incorrect or impossible relationships between objects, it would be useful to illustrate how these issues are addressed to ensure coherent generation results.

**Questions:**

Providing feedback on questions discussed in the weaknesses will be helpful.

---

### Official Review · Reviewer_834L · 2024-11-03

**Soundness:** 3
**Presentation:** 3
**Contribution:** 3
**Rating:** 5
**Confidence:** 3

**Summary:**

The authors propose "Scene Language," a pipeline for producing -- and later rendering -- a compositional scene representation from a text prompt or image. In contrast with GraphDreamer, which is compared against as an "exemplar approach," the authors task a language model with generating a precise, text-based code representation to define scene layout. The authors experiment with the application of multiple "rendering modules" to realize the code-based representations into explicit scenes, requiring only minor prompting modifications. To evaluate their method, the authors conduct a perceptual study measuring prompt alignment and counting accuracy.

**Strengths:**

1. The paper is well-written.
2. The pipeline outputs are visually appealing and appear generally well-aligned with the prompt texts.
3. The task is interesting and relevant to the ICLR community.

**Weaknesses:**

1. The authors make several unsupported claims, detailed below:

    a. "In summary, our contributions are as follows... Empirical results on text- and image-conditioned scene generation and editing tasks." Neither image-conditioned scene generation nor editing were empirically evaluated. The only such evaluation involved text-conditioned scene generation: "We perform a user study to compare with prior methods on the text-conditioned 3D generation task and report the percentages of user preferences for prompt alignment."

    b. "Compared with existing methods, our Scene Language produces 3D and 4D scenes with significantly higher fidelity, preserves complex scene structures, and enables easy and precise editing." 4D generation was not included in any evaluations or method comparisons.

    c. "Together, this forms a robust, fully automated system for high-quality 3D and 4D scene generation." The authors neither evaluate pipeline robustness nor include any discussion of it.
2. The authors' choice of evaluations raises concerns:

    a. Rather than evaluate their pipeline on an existing setting, the authors opt to pick their own evaluation set of nine prompts, each of which includes a number ("8-layer", "5x5", "four", etc.). On this set, the authors measure "counting accuracy (0 for inaccurate and 1 for accurate)". They "compare with GraphDreamer (Gao et al., 2024) as an exemplar approach," but note that when the GraphDreamer "raw scene graph output contains too many objects, (they) rerun the graph generation and add 'The maximum number of objects is three.' in text prompt to avoid reaching memory limitation during optimization." This casts doubt on the significance of the results. Is the proposed method generally applicable, or does it only excel in counting-related scenarios with four or more objects?

    b. The authors display an image-to-scene comparison with GraphDreamer in Figure 6 and remark "Compared with our method, which preserves both structure and visual content from input images, GraphDreamer only reconstructs semantics from input images and leaves out entity poses and identities, due to the information loss in the intermediate scene graph representation." However, it is not clear why GraphDreamer (which is inherently semantic) was chosen for this comparison when the authors could have evaluated against a monocular reconstruction method.
3. The authors prominently feature the use of embeddings as a contribution of their work. However, except for serving as UUIDs, they seem only to be meaningfully employed in the image-to-scene task where a segmentation model is used to localize regions to apply textual inversion to. The authors' characterization of the embeddings as describing "the attributes and identity of the output entity, like a specific color of a 'pawn'." is an interesting idea but does not appear to align with their use in the paper: the black-and-white chess pieces clearly do not share shapes. It does not seem as though the authors have taken the idea far enough, and as it stands, could be removed from the pipeline to no discernible effect.

**Questions:**

1. Regarding the image-to-scene task, since the LLM is already employed to describe the objects in the scene in order to prompt SAM, the textual-inversion step seems unnecessary or at the least deserving of an ablation study. Have the authors evaluated its effectiveness over using only LLM descriptions?
2. The Gaussians and Minecraft lecture-hall samples in Figure 9 seem to have very similar layouts. This is confusing given that the renderers require different scene-generation prompts ("To prompt LM to generate Minecraft-compatible outputs, we remove rotation matrix and reflection matrix from the system prompt in Appendix E.1 and change the function header for primitive call to the follows:") and so must be the results of distinct LLM calls. Do the authors have intuition as to why the layouts appear so similar? How much layout variability is observed between successive calls?
3. Is any differentiable rendering done with Mitsuba? It appears to be employed only as a generic physically based renderer and it's unclear why it was chosen over a more standard graphics engine.

Addl. Comments:
1. The use of Lisp-like syntax in Figure 2 is confusing as the authors "prompt LMs to generate a Python program." The authors should consider using the actual syntax throughout to improve clarity.
2. The "chessboard at game start" is incorrectly configured as the queens are not on their color.
3. The staircase code edit in Figure 5 displays the wrong values.

---

### Official Review · Reviewer_6kJL · 2024-11-04

**Soundness:** 3
**Presentation:** 3
**Contribution:** 3
**Rating:** 6
**Confidence:** 4

**Summary:**

This paper utilizes the formal structure of the LISP language to define scenarios, allowing for precise expression of scenes and benefiting tasks such as generation and editing.

**Strengths:**

The value of accurate coding is self-evident. The performance in tasks is remarkable and addresses issues where traditional language instructions are difficult to follow effectively.

**Weaknesses:**

- However, overly precise definitions can significantly reduce flexibility. For example, "a child with golden hair is gazing out the window from the desk" is a very common and simple description in traditional language, but defining it using LISP is extremely challenging. Although embedding methods exist, I doubt whether this would completely degrade into pure embeddings, failing to effectively leverage the advantages of formal languages.

- The downside of using a LISP-like language for definitions is the extremely long textual information. This not only significantly increases memory consumption but also makes training more difficult. While the article uses a very clever method to circumvent this issue, it could lead to difficulties in subsequent work.

- Direct comparisons with previous work are unfair because this article is more akin to a combination of 3D assets, while previous work involves direct generation. The former relies on the latter. Essentially, they are different tasks, so a fair comparison should be: GPT-4 or a rule-based model directly decomposing prompts and generating combinations.

**Questions:**

Refer to weaknesses.

---

### Author Response · Authors · 2024-12-02
**General Response (Cont'd)**

**Q4 - Limitations (`834L`, `fKvz`, `K12Q`)**

We identify two limitations of the current inference pipeline. First, it is sensitive to prompts. For text-conditioned tasks, minor variations in textual scene descriptions can lead to large quality differences in the output. Second, a typical failure mode is the image parsing error. For image-conditioned tasks, input images are parsed with the backbone visual language model. With the same input image, parsing results have high variance across multiple inference runs. Failure case results are updated on the [project page](https://sclg-page.github.io/#failure).

While the current paper provides a viable inference method for the proposed representation that leverages the commonsense knowledge and code-writing capability of LMs for scene generation and editing tasks, we fully agree with reviewers `834L`, `fKvz`, and `K12Q` that addressing the weaknesses inherited from LMs would provide further improvement in robustness and output quality for downstream tasks. We leave these as exciting future directions to improve the inference of the Scene Language, and emphasize that the representation itself is the focus and the core contribution of this paper.


___
References
1. Gao, G., Liu, W., Chen, A., Geiger, A., & Schölkopf, B. (2024). GraphDreamer: Compositional 3D scene synthesis from scene graphs. In Proceedings of the IEEE/CVF Conference on Computer Vision and Pattern Recognition (pp. 21295-21304).
2. Zhang, R., Isola, P., Efros, A. A., Shechtman, E., & Wang, O. (2018). The unreasonable effectiveness of deep features as a perceptual metric. In Proceedings of the IEEE conference on computer vision and pattern recognition (pp. 586-595).
3. Bahmani, S., Skorokhodov, I., Rong, V., Wetzstein, G., Guibas, L., Wonka, P., ... & Lindell, D. B. (2024). 4D-fy: Text-to-4D generation using hybrid score distillation sampling. In Proceedings of the IEEE/CVF Conference on Computer Vision and Pattern Recognition (pp. 7996-8006).
4. Ilharco, G., Wortsman, M., Carlini, N., Taori, R., Dave, A., Shankar, V., Namkoong, H., Miller, J., Hajishirzi, H., Farhadi, A., & Schmidt, L. (2021). OpenCLIP (0.1). Zenodo.
5. Huang, Z., He, Y., Yu, J., Zhang, F., Si, C., Jiang, Y., ... & Liu, Z. (2024). VBench: Comprehensive benchmark suite for video generative models. In Proceedings of the IEEE/CVF Conference on Computer Vision and Pattern Recognition (pp. 21807-21818).
6. Teed, Z., & Deng, J. (2020). RAFT: Recurrent all-pairs field transforms for optical flow. In Computer Vision–ECCV 2020: 16th European Conference, Glasgow, UK, August 23–28, 2020, Proceedings, Part II 16 (pp. 402-419). Springer International Publishing.

---

### Author Response · Authors · 2024-12-02
**General Response**

We sincerely thank all reviewers for their thoughtful responses. We are glad that reviewers found our representation effective in addressing issues with natural language instructions (`6kJL`), and our task interesting (`834L`) and practically important (`fKvz`), with visually appealing results (`834L`, `K12Q`). Reviewers raise valuable concerns and feedback, which greatly help improve this work. We address some shared concerns below and will include all additional results and discussions in future revisions of this work for the next venue.

**Q1 - Quantitative evaluation (`834L`, `fKvz`, `K12Q`)**

To evaluate the proposed inference method more rigorously, we provide additional quantitative evaluations described below.

For image-conditioned generation, we clarify that the focus of this task is generating scenes coherent with the semantics and structures parsed from input images. This is most similar to the “inverse semantics” task from GraphDreamer [1] and differs from the monocular 3D reconstruction task, which aims to reconstruct the 3D geometry and appearance whose 2D projection has perfect, pixel-wise alignment with input images. This is why we compared with GraphDreamer in the paper as a baseline (`834L`).

We report the LPIPS [2] scores between an input image and an output rendering, averaged over the three scenes from Figure 6 with 20 viewpoints each below for a quantitative comparison.

| Methods          | LPIPS [2] (↓) |
| ---------------- | ------------- |
| GraphDreamer [1] | 0.811         |
| Ours             | 0.681         |

For text-conditioned 4D generation, we compare with 4D-fy [3] as an exemplar text-to-4D generation method. We measure the prompt alignment of output dynamic renderings using CLIP similarity [4] between rendered frames and input texts, and the motion magnitude using dynamic degrees [5] computed using RAFT [6] optical flow estimation between neighboring frames. Results averaged across the three prompts from Figure 7 are shown below. We observed that 4D-fy results have a higher per-frame alignment score than ours but present very small dynamic motions. See the [project page](https://sclg-page.github.io/#4d) for qualitative comparisons.

| Methods   | CLIP Similarity [4] (↑) | Dynamic Degrees [5] (↑) |
| --------- | ----------------------- | ----------------------- |
| 4D-fy [3] | 0.352                   | 0.2%                    |
| Ours      | 0.341                   | 5.9%                    |



**Q2 - Scene diversity and complexity and occlusion handling (`834L`, `fKvz`, `K12Q`)**

The [project page](https://sclg-page.github.io/#renderers) shows both indoor, tabletop scenes (e.g., “A monitor, a keyboard, a mouse, a metal photo frame, and a plant on a wooden desk top”) and outdoor scenes (e.g., “A scene inspired by The Angelus, Millet”). Representing spatial relations of entities in 3D, instead of 2D, by parameterizing 3D poses of entities in the proposed representation is exactly why occlusion can be well-handled in these scenes. For image-conditioned tasks, when input images contain occlusions (`K12Q`), e.g., in the coke-can scene in Figure 6, we observed that LMs still infer a Scene Language program with all 6 cans, including the occluded ones.

**Q3 - The role of embeddings (`834L`)**

While texts provide an easy user interface for scene generation, image prompting as an additional interface allows users to specify further intentions that could be hard to describe with natural language alone. In this work, neural embeddings indeed serve as UUIDs, as noted by reviewer `834L`, and in addition, increase the expressivity of the representation to encode visual details from image prompts.

Practically, embeddings are crucial in two tasks studied in the paper: image-conditioned scene generation and image-conditioned editing.
* For image-conditioned scene generation, embeddings enable better visual content preservation than a scene-graph-based approach, GraphDreamer, as quantified in the first table above.
* For image-conditioned style-transfer tasks, the role of embeddings is twofold. First, they allow targeted regional or global edits. Specifically, they allow for edits targeting partial or full scenes, as shown in the [project page](https://sclg-page.github.io/#text-cond), by updating embeddings of the statues or the full scene (composed from the statues and the base), respectively. Second, they allow for preserving style details from input images.

---

### Note · Authors · 2024-11-15

I have read and agree with the venue's withdrawal policy on behalf of myself and my co-authors.

---

> ### Note · Program_Chairs · 2024-12-02
>
> **Comment:**
>
> Withdrawal reversed temporarily.
>
> **Revert Withdrawal Confirmation:**
>
> We approve the reversion of withdrawn submission.

---

### Note · Authors · 2024-12-02

**Comment:**

Requested by authors.

**Withdrawal Confirmation:**

I have read and agree with the venue's withdrawal policy on behalf of myself and my co-authors.